# Lack of TYK2 signaling enhances host resistance to *Candida albicans* skin infection

Sara Miranda[1], Caroline Lassnig[1,2], Kristina Schmidhofer[1], Hrönn Kjartansdottir [1], Claus Vogl [1], Simone Tangermann[3], Irina Tsymala[4,5], Verena Babl[1], Mathias Müller [1,2], Karl Kuchler [4,5] & Birgit Strobl [1] ✉

*Candida albicans* is the most common human fungal pathogen, causing diseases ranging from local to life-threating systemic infections. Tyrosine kinase 2 (TYK2), a crucial mediator in several cytokine signaling pathways, has been associated with protective functions in various microbial infections. However, its specific contribution in the immune response to fungal infections has remained elusive. In this study, we show that mice lacking TYK2 or its enzymatic activity exhibit enhanced resistance to *C. albicans* skin infections, limiting fungal spread and accelerating wound healing. Impaired TYK2-signaling prompted the formation of a distinctive layer of necrotic neutrophils around the fungal pathogens. Transcriptomic analysis revealed TYK2's pivotal role in regulating interferon-inducible genes in neutrophils, thereby impacting their antifungal capacity during infection. Furthermore, we show that TYK2-dependent interferon-gamma (IFNγ) production contributes to fungal dissemination from the skin to the kidneys. Our study uncovers a hitherto unrecognized detrimental role of TYK2 in cutaneous *C. albicans* infections.

*Candida albicans* is the leading cause of opportunistic fungal infections in humans[1]. In healthy individuals, *C. albicans* resides as a commensal in mucosal tissues, including the oral, vaginal, gastrointestinal tracts, and the skin[2,3]. If the integrity of the barrier tissues is perturbed or the host immune system is impaired, asymptomatic colonization by *C. albicans* can progress to infections that range from superficial to life-threatening systemic infections[4]. A balanced immune defense is essential to tolerate harmless commensal colonization but also to prevent fungal invasion by promoting efficient pathogen clearance when necessary. Efficient protection against candidiasis requires cooperative activity of the innate and adaptive branches of the immune system[3,5]. Together with epithelial cells, innate immune cells constitute the first line of defense against the pathogen. Tissue-resident macrophages recruit other immune cells, such as blood monocytes and neutrophils, to the site of infection[6]. In the skin and mucosal tissues, γδ T cells also contribute to neutrophil recruitment via IL-17 production[7–10]. Neutrophils rapidly reach the infected tissues

and eliminate pathogens by intra- and extra-cellular killing mechanisms[11]. Neutrophils are short-lived cells that can undergo many different types of cell death after performing their effector functions[12]. Dead neutrophils are removed by macrophages and dendritic cells (DCs), an essential step to resolve inflammation and restore tissue homeostasis[11,13]. DCs recognize the pathogen and present fungal antigens to T cells, triggering an adaptive immune response that results in long-lasting protection against the fungal pathogen[5].

Tyrosine kinase 2 (TYK2) is a member of the Janus kinase (JAK) family of non-receptor protein tyrosine kinases. TYK2 is crucial in signal transduction induced by a variety of cytokines, including type I interferons (IFN-I), interleukin (IL)−12, IL-22, IL-23 and, most likely in a cell type- or differentiation-dependent manner, IL-10. Absence of TYK2 in humans and mice causes high susceptibility to bacterial and viral infection, which is mainly associated with impaired production of IFNγ in response to IL-12 and defective IFN-I signaling[14–16]. Moreover, homozygosity for common TYK2 missense variants, including variants

[1]Centre of Biological Sciences, University of Veterinary Medicine Vienna, Vienna, Austria. [2]Vetbiomodels, University of Veterinary Medicine Vienna, Vienna, Austria. [3]Centre of Pathobiology, University of Veterinary Medicine Vienna, Vienna, Austria. [4]Max Perutz Labs, Vienna Biocenter Campus (VBC), Vienna, Austria. [5]Center for Medical Biochemistry, Medical University of Vienna, Vienna, Austria. ✉e-mail: birgit.strobl@vetmeduni.ac.at

encoding catalytically inactive TYK2, impairs IL-23-dependent IFNγ production and predisposes patients to infections with *Mycobacterium tuberculosis*[17,18]. In contrast to its protective functions during infectious diseases, TYK2 promotes autoimmune and inflammatory diseases in humans and mice. Single nucleotide polymorphisms in the *TYK2* locus are associated with a wide-range of autoimmune diseases, including psoriasis, rheumatoid arthritis, multiple sclerosis, and Crohn's disease[19]. Recent years have witnessed a tremendous increase in the exploration of TYK2 inhibition as a therapeutic strategy for treating such diseases[20]. The first selective TYK2 inhibitor, Deucravacitinib, has recently been approved for the treatment of moderate to severe plaque psoriasis[21] and numerous clinical trials for additional diseases and with other TYK2 inhibitors are ongoing. It is worth noting, however, that TYK2 possesses scaffolding functions that are independent of its enzymatic activity[22–24]. Therefore, it is extremely important to gain deeper insights into TYK2 kinase-dependent and independent functions in different pathophysiological conditions. Moreover, there are still gaps in our understanding of how TYK2 balances host-protective and disease-promoting immunity in vivo, in particular in response to fungal infections and at tissue barriers.

In the present work, we analyzed the role of TYK2 in the immune response against cutaneous candidiasis using gene-targeted mouse models, including complete and conditional *Tyk2* knockout mice as well as mice expressing a kinase-inactive version of TYK2. We observed that mice lacking TYK2 or its catalytic activity are more resistant to intradermal infection with *C. albicans* than wildtype (*WT*) mice. We show that in the absence of catalytically active TYK2, *C. albicans* remains locally contained at the infection site and does not invade deep skin layers or disseminate to kidneys. The better containment of *C. albicans* correlates with the formation of a dense cell layer around the wound, consisting primarily of necrotic neutrophils that seem to mechanically impair fungal dissemination. We furthermore provide collective genetic evidence that TYK2-driven production of IFNγ by innate-like T cells facilitates *C. albicans* invasion and dissemination to kidneys. Taken together, our results unravel a detrimental role of TYK2 in cutaneous candidiasis and provide insights into the immune response to local fungal infections. Our findings that the detrimental function of TYK2 is dependent on its kinase activity paves the way for exploring TYK2 inhibitors as potential therapeutic options.

## Results

### Absence of catalytically active TYK2 increases host resistance to *C. albicans* intradermal infection

To investigate enzymatic and scaffolding functions of TYK2 in the immune response to local fungal infections, we used *Tyk2*-deficient mice (*Tyk2-/-*), mice expressing kinase-inactive TYK2 (*Tyk2^K923E*)[25], and *WT* mice in a model of *C. albicans* skin infection. Mice were intradermally infected with *C. albicans* and fungal burdens in the skin and the kidneys were assessed on day 2 and day 4 post-infection (p.i) (Fig. 1a). *Tyk2-/-* and *Tyk2^K923E* mice showed reduced fungal load in the skin in comparison to *WT* mice (Fig. 1b, Supplementary Fig. S1a). More strikingly, *Tyk2-/-* and *Tyk2^K923E* mice showed almost no dissemination to kidneys (Fig. 1c, Supplementary Fig. S1b). Alterations in the blood cell composition after infection were evaluated to characterize the systemic response to the local infection. We observed a significant increase in the frequency of granulocytes at the expense of lymphocytes in the white blood cell compartment of *WT* mice at day 4 p.i (Fig. 1d, Supplementary Fig. S1c). In the absence of TYK2 or its kinase activity, these infection-induced changes were less pronounced than in control mice (Fig. 1d, Supplementary Fig. S1c). No significant alterations were observed in the red blood cell compartment and the total white blood cell counts (Supplementary Fig. S1d). To understand if the absence of TYK2 or its kinase activity influences the fungal distribution in the skin, a Gomori Methenamine-Silver (GMS) staining was performed on skin sections. In *WT* mice, *C. albicans* spread throughout the

skin, whereas fungi remained more contained at the infection site in *Tyk2-/-* and *Tyk2^K923E* mice, which only showed a few *C. albicans* hyphae penetrating the deeper layer of the skin (Fig. 1e).

Intradermal infection with *C. albicans* leads to the development of wounds that typically manifest as a nodule within the first 24 to 48 h after the infection. The initial nodule is accompanied by ulceration and redness, which eventually results in the formation of a crust during the healing process[26]. To assess the impact of TYK2 on disease progression, wounds were visually monitored for 24 days in *Tyk2-/-*, *Tyk2^K923E*, and *WT* mice (Fig. 1f). Mice were considered healed when no nodule, ulceration, redness, or crusting was detectable. In line with previous studies[27], it took about 22 days for all *WT* mice to be fully healed. *Tyk2-/-* and *Tyk2^K923E* mice were completely healed earlier than *WT* mice (Fig. 1g). Moreover, *Tyk2-/-* and *Tyk2^K923E* mice recovered faster from the infection-induced body weight loss than control mice (Supplementary Fig. S1e).

Overall, these results demonstrate that TYK2 signaling promotes local *C. albicans* growth and fungal dissemination to kidneys and delays the recovery from the infection.

### Lack of TYK2 or its catalytic activity promotes the formation of a necrotic neutrophil layer surrounding *C. albicans* in the skin

We next analyzed innate immune cell infiltration into the skin by immunohistochemistry (IHC) using the NIMP-R14 antibody, which recognizes neutrophils and inflammatory monocytes (i.e., Ly6C+ and Ly6G+ cells). As expected, we found massive infiltration by Ly6C/G+ cells at day 1, day 2, and day 4 p.i. (Fig. 2a, Supplementary Fig. S2a). There were no gross differences between *WT* and TYK2-mutant mice (Fig. 2a, Supplementary Fig. S2a). In line with the IHC results, we did not observe differences in neutrophil and monocyte infiltration between *WT, Tyk2-/-* and *Tyk2^K923E* mice using flow cytometry (Fig. 2b–e and Supplementary Fig. S2b, c). Moreover, the number and frequency of macrophages at the infection site was similar in *WT, Tyk2-/-* and *Tyk2^K923E* mice (Supplementary Fig. S2d).

Despite being terminally differentiated cells, activation of neutrophils reactivates cell cycle-associated markers such as Ki-67[28]. A large proportion of neutrophils surrounding *C. albicans* were Ki-67+ in mice of all genotypes on day 1 p.i. (Fig. 2g). In contrast, a distinct layer of Ki-67-negative cells immediately adjacent to the fungi was visible in skin sections from *Tyk2-/-* and *Tyk2^K923E* mice but not from *WT* mice on day 2 and day 4 p.i. (Fig. 2h, Supplementary Fig. S2g). A similar pattern was observed when skin sections were stained with an anti-myeloperoxidase (MPO) antibody, which reacts with activated neutrophils (Supplementary Fig. 2h). Pathological examination revealed that the majority of Ki-67- and MPO-negative cells immediately adjacent to *C. albicans* displayed morphological signs of necrosis, characterized by loss of cellular detail, indistinct cell borders, karyopyknosis, karyorrhexis, karyolysis, and nuclear smearing, suggesting that the absence of TYK2 or its enzymatic activity increases neutrophil necrosis at the infection site.

Collectively, these data indicate that the absence of TYK2 does not affect neutrophil recruitment to infection sites but causes the formation of a necrotic neutrophil "barrier" that may trap the fungal pathogen.

### Neutrophil depletion abolishes the differences in *C. albicans* dissemination between *WT* and *Tyk2-/-* mice

To test whether neutrophils are involved in controlling fungal dissemination, we injected *WT* and *Tyk2-/-* mice with an anti-Ly6G antibody one day before *C. albicans* infection (Fig. 3a). Efficient depletion of neutrophils in the blood and skin of *WT* and *Tyk2-/-* mice was confirmed with flow cytometry (Fig. 3b, c, Supplementary Fig. S3c, d) and IHC for NIMP-R14 (Fig. 3d) at day 2 post infection. Neutrophil depletion resulted in increased *C. albicans* penetrance into deeper skin layers in both *WT* and *Tyk2-/-* animals (Fig. 3e) and increased fungal dissemination into the kidneys of *Tyk2-/-* mice to levels observed in *WT* mice

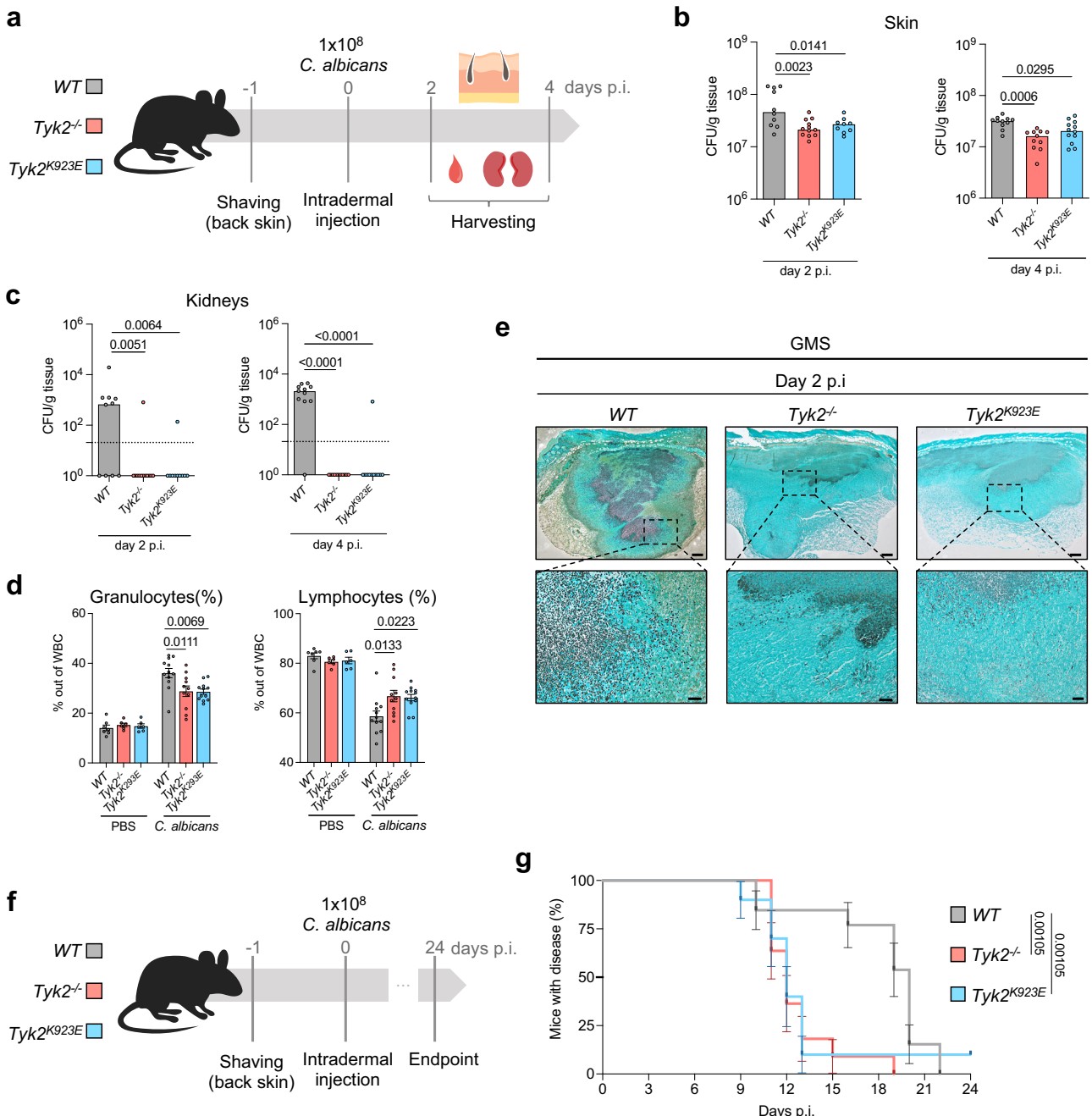

**Fig. 1 | Absence of enzymatically active TYK2 results in less severe candidiasis.**
**a** *WT*, *Tyk2⁻/⁻*, and *Tyk2^K923E* mice were intradermally (i.d) injected in the back skin with $1 \times 10^8$ CFU of *C. albicans* or PBS (control). Fungal load in the skin (**b**) and in the kidneys (**c**) was measured on days 2 and 4 post-infection (p.i). For each time-point, pooled data from 2 independent experiments are shown, and median values are given; n: biological replicates; the dotted line indicates the assay detection limit (**b**, **c**). Skin day 2: n = 10 (*WT*), n = 12 (*Tyk2⁻/⁻*) and n = 9 (*Tyk2^K923E*); skin day 4: n = 11 (*WT*, *Tyk2⁻/⁻*) and n = 12 (*Tyk2^K923E*) (**b**); kidney day 2: n = 10 (*WT*, *Tyk2^K923E*) and n = 12 (*Tyk2⁻/⁻*); kidney day 4: n = 11/genotype (**c**). **d** Blood cell composition on day 4 p.i was determined with a Vet ABC analyzer. The percentage of granulocytes and lymphocytes out of total white blood cells (WBC) is shown. Pooled data from 2 independent experiments are shown. Mean values ± SEM are given; PBS: n = 7 (*WT*) and n = 6 (*Tyk2⁻/⁻*, *Tyk2^K923E*); *C. albicans*: n = 12 (*WT, Tyk2^K923E*) and n = 11 (*Tyk2⁻/⁻*);

n: biological replicates. **e** Representative pictures of the infected skin on day 2 p.i. A Gomori Methenamine-Silver (GMS) staining of the skin sections is shown. Scale bar: 200 μm (top), 50 μm (bottom). Data are representative for sections from 7 mice per genotype. **f** *WT*, *Tyk2⁻/⁻* and *Tyk2^K923E* mice were infected as described above and wounds were visually monitored for 25 days. **g** The percentage of mice with wounds is shown. Pooled data from 2 independent experiments are shown. Error bars indicate the standard error (SE), centers for the error bars indicate mean values; n = 13 (*WT*), n = 11 (*Tyk2⁻/⁻*) and n = 10 (*Tyk2^K923E*); n: biological replicates. Statistical analysis was conducted using One-way ANOVA followed by Tukey's multiple comparison test (**b**–**d**) and the Log-rank (Mantel-Cox) test (**g**). Statistical significance is only given for the comparison between the genotypes (**b**–**d**, **g**). Source data are provided as a Source Data file.

(Fig. 3f, Supplementary Fig. S3g). Notably, neutrophil depletion also increased fungal load in the kidneys from *WT* mice, albeit to a substantially lower extent than in *Tyk2⁻/⁻* mice.

Overall, these results indicate that the increased control of *C. albicans* invasion and dissemination to kidneys in *Tyk2⁻/⁻* compared to *WT* mice depends on the presence of neutrophils.

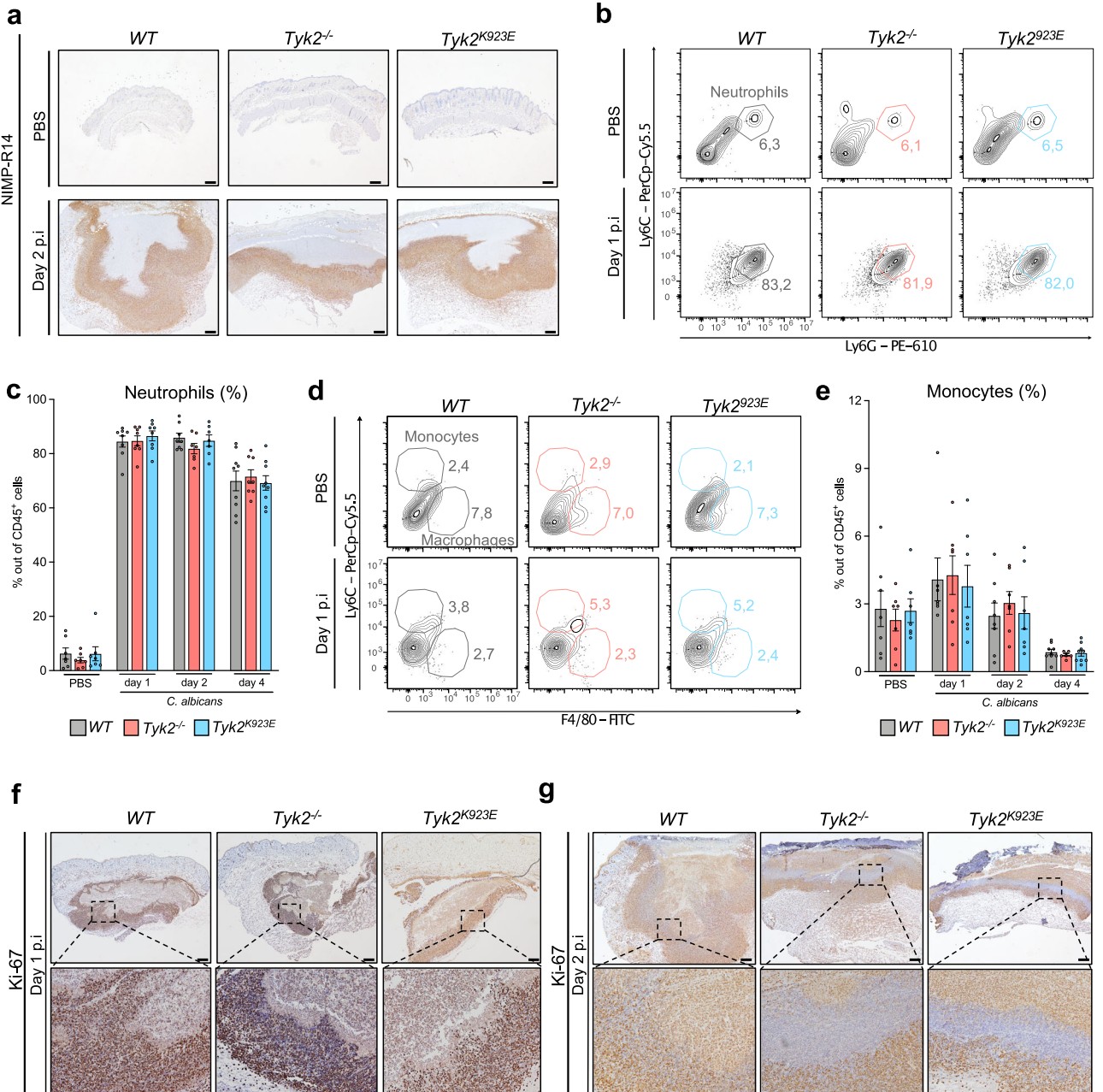

**Fig. 2 | *C. albicans* infection induces TYK2-independent infiltration of neutrophils, which form a Ki-67-negative cell layer around the pathogen in TYK2 mutants but not in *WT* mice. a** Representative IHC pictures from the infected skin of *WT*, *Tyk2⁻/⁻*, and *Tyk2^K923E* mice on day 2 p.i. Sections were stained with an anti-NIMP-R14 antibody (which stains Ly6C⁺ and Ly6G⁺ cells). Scale bar: 200 μm. Data are representative of sections from 3 mice per genotype (PBS) and from 7 mice per genotype (day 2). **b**−**e** The percentages of skin-infiltrating neutrophils (gated as CD45⁺CD11b⁺Ly6C⁺Ly6G⁺ cells) (**b**, **c**) and monocytes (gated as CD45⁺CD11b⁺ Ly6G⁻Ly6C^high F4/80⁻ cells) (**d**, **e**) out of CD45⁺CD11b⁺ cells were determined by flow cytometry analysis on days 1, 2 and 4 p.i. Representative contour plots (**b**, **d**) and pooled data from 2 independent experiments for each time-point (**c**, **e**) are shown.

PBS: n = 7/genotype (**c**, **e**), *C. albicans* day 1: n = 8/genotype (**c**) and n = 7/genotype (**e**); *C. albicans* day 2: n = 8 (*WT*) and n = 7 (*Tyk2⁻/⁻*, *Tyk2^K923E*) (**c**, **e**); *C. albicans* day 4: n = 9 (*WT*, *Tyk2^K923E*), n = 8 (*Tyk2⁻/⁻*) (**c**) and n = 9 (WT, *Tyk2^K923E*) and n = 7 (*Tyk2⁻/⁻*) (**e**); Mean values ± SEM are given; n: biological replicates. Statistical analysis was conducted using One-way ANOVA followed by Tukey's multiple comparison test; statistical significance is only given for the comparison between the genotypes (**c**, **e**). Representative IHC pictures from the infected skin on day 1 (**f**) and 2 (**g**) p.i. Sections were stained with an anti-Ki-67 antibody. Scale bar: 200 μm (top), 50 μm (bottom). Data are representative of sections from 5 mice per genotype (**f**) and from 7 mice per genotype (**g**). Source data are provided as a Source Data file.

## Loss of TYK2 or its kinase activity impairs IFN-inducible gene expression in skin-infiltrating neutrophils

To better understand the impact of TYK2 on the functionality of skin-infiltrating neutrophils, we performed RNA-sequencing of sorted neutrophils. To obtain sufficient and equal numbers of viable neutrophils, we isolated myeloid cells (CD45⁺CD3⁻CD19⁻NK1.1⁻) from

infected skin at day 1 p.i from *WT, Tyk2⁻/⁻* and *Tyk2^K923E* mice (Fig. 4a). The sorted population consisted of around 80% neutrophils with more than 80% viability (Supplementary Fig. S2c, f). A total of 769 and 532 differentially expressed genes (DEGs) were found between *Tyk2⁻/⁻* and *WT* cells and *Tyk2^K923E* and *WT* cells, respectively (p_adj < 0.05) (Fig. 4b, Supplementary Data), whereas only 18 genes were differentially

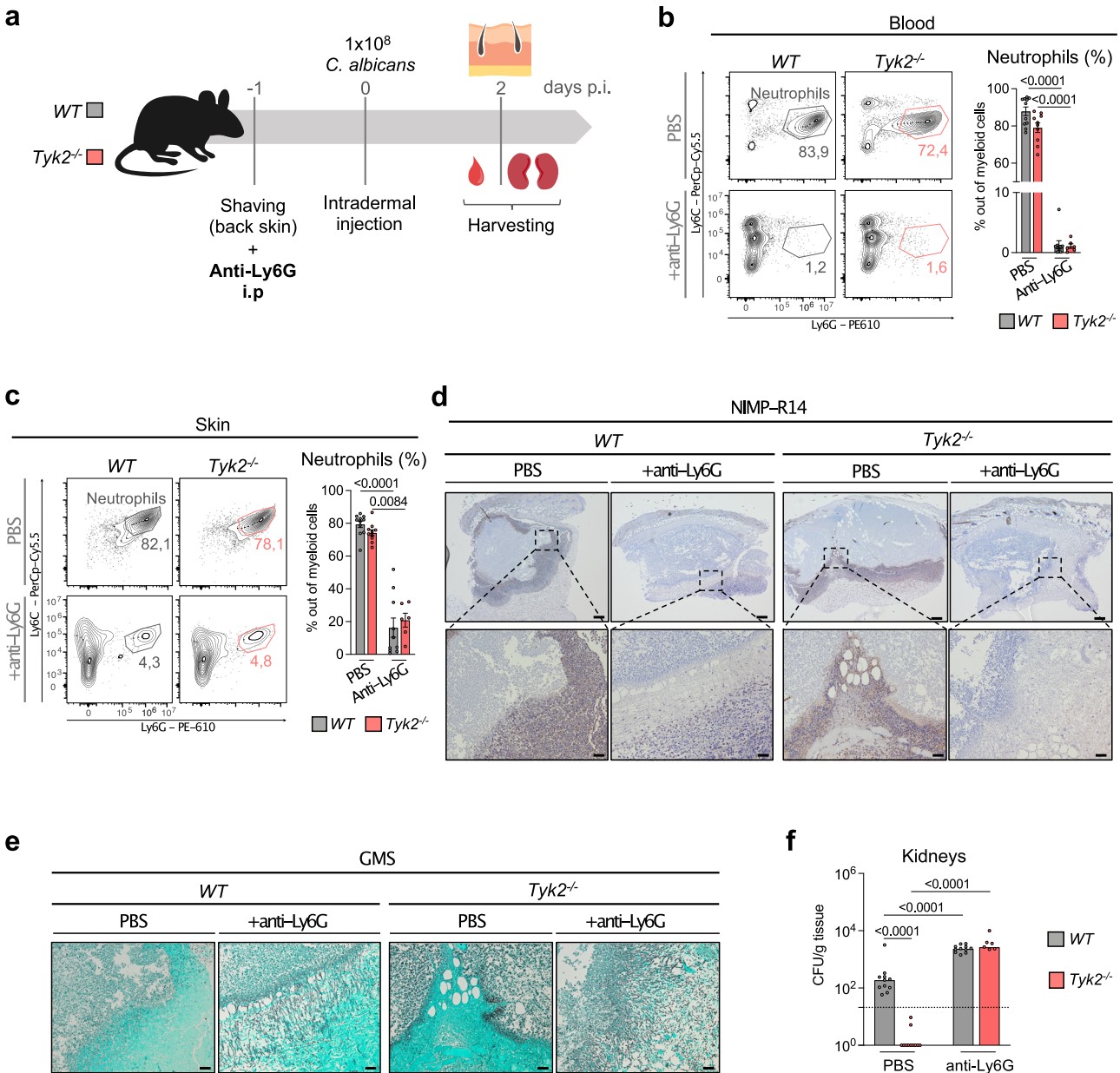

**Fig. 3 | Depletion of neutrophils results in increased fungal dissemination to the kidneys. a** *WT* and *Tyk2⁻ᐟ⁻* mice were intraperitoneally (i.p) injected with an antibody against Ly6G (anti-Ly6G) or PBS (control) one day before being intra-dermally (i.d) injected in the back skin with 1×10⁸ CFU of *C. albicans*. The percen-tages of neutrophils (gated as CD45⁺Ly6C⁺Ly6G⁺ cells) out of CD45⁺ cells in the blood (**b**) and the skin (**c**) were determined by flow cytometry analysis on day 2 p.i. Representative contour plots and pooled data from 2 independent experiments are shown and mean values ± SEM are given; PBS: n = 11 (*WT*), n = 10 (*Tyk2⁻ᐟ⁻*); anti-Ly6G: n = 10 (*WT*) and n = 6 (*Tyk2⁻ᐟ⁻*); n: biological replicates (**b**, **c**). **d** Representative IHC pictures from the infected skin on day 2 p.i. Sections were stained with an anti-NIMP-R14 antibody (which stains Ly6C⁺ and Ly6G⁺ cells). PBS: Data are repre-sentative for sections from 8 (*WT*) and 11 (*Tyk2⁻ᐟ⁻*) mice. Anti-Ly6G: Data are

representative for sections from 9 (*WT*) and 6 (*Tyk2⁻ᐟ⁻*) mice. Scale bar: 200 µm (top), 50 µm (bottom). **e** Representative pictures of the infected skin on day 2 p.i. A GMS staining of the skin sections is shown. PBS: Data are representative for sections from 8 (*WT*) and 11 (*Tyk2⁻ᐟ⁻*) mice. Anti-Ly6G: Data are representative for sections from 9 (*WT*) and 6 (*Tyk2⁻ᐟ⁻*) mice. Scale bar: 50 µm. **f** Fungal load in the kidneys of infected mice previously treated with anti-Ly6G antibody or PBS was measured on day 2 p.i. Pooled data from 2 independent experiments are shown and median values are given; PBS: n = 11/genotype; anti-Ly6G: n = 11 (*WT*) and n = 6 (*Tyk2⁻ᐟ⁻*); ns: not significant; n: biological replicates. The dotted line indicates the assay detection limit. Statistical analysis was conducted using One-way ANOVA followed by Tukey's multiple comparison test and statistical significance is only given for the compar-ison between the genotypes (**b**, **c**, **f**). Source data are provided as a Source Data file.

expressed between *Tyk2⁻ᐟ⁻* and *Tyk2^K923E* cells (Supplementary Fig. S4a). The top downregulated genes in *Tyk2⁻ᐟ⁻* and *Tyk2^K923E* cells included many interferon-stimulated genes (ISGs), suggesting that skin-infiltrating neutrophils mount a defective response to IFNs in the absence of catalytically active TYK2 (Fig. 4c). In line, pathway analysis revealed a strong enrichment of genes annotated to "interferon alpha/beta signaling", "interferon signaling" and "cytokine signaling in the

immune system" (Fig. 4d). Among pathways upregulated in the absence of TYK2, genes annotated to "neutrophil degranulation" and "translation" displayed the strongest enrichment (Fig. 4d).

The manual inspection of gene lists previously associated with various biological processes in neutrophils[29] did not reveal any sig-nificant differences in the expression of genes related to (i) neutrophil migration, such as *Cxcr2, S100a8, Vcam1, Cd9, C3ar1, Cx3cr1* or *Icam1*,

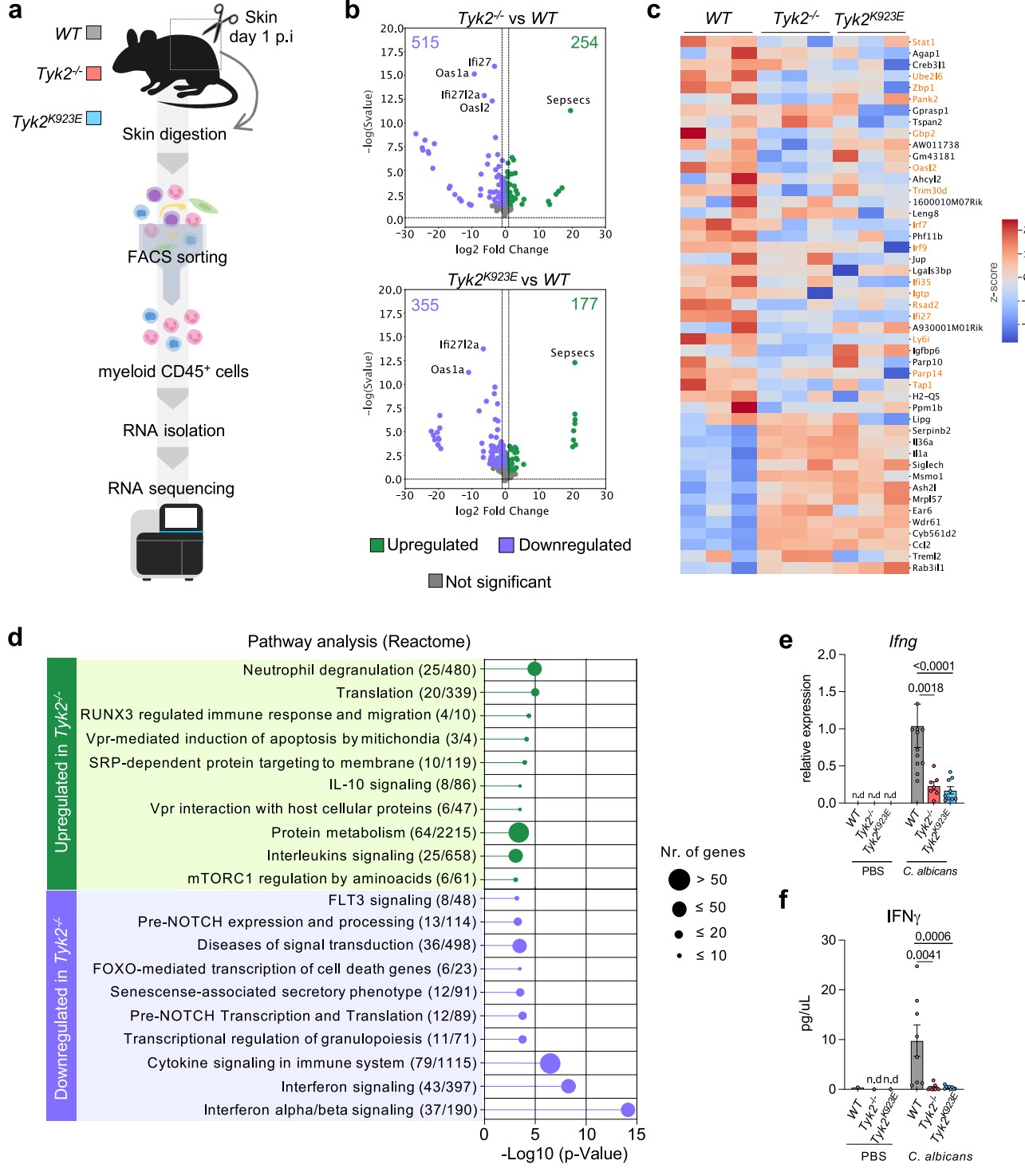

**Fig. 4 | Absence of TYK2 or its kinase activity alters the transcriptome of skin-infiltrating myeloid cells. a** *WT*, *Tyk2⁻/⁻* and *Tyk2^K923E* mice were infected as described in the legend to Fig. 1. RNA from skin-sorted CD45⁺CD3⁻CD19⁻NK1.1⁻ myeloid cells was isolated for subsequent RNA-sequencing. **b** Volcano plots showing the differentially expressed genes between *Tyk2⁻/⁻* and *WT* and *Tyk2^K923E* and *WT* cells. Genes were identified from DESeq2-normalized read counts of genes with a threshold of padj < 0.05 between *Tyk2⁻/⁻* and *WT* and *Tyk2^K923E* and *WT* cells. Data are from one experiment (n = 3 mice per genotype). **c** Heatmap showing the differentially expressed genes between WT and TYK2-mutant cells with an absolute log2-fold change > 1 (padj <0.05). IFN-stimulated genes (ISGs) are highlighted in orange. **d** Pathway analysis of deregulated pathways between *WT* and ^Tyk2⁻/⁻ cells was performed using Reactome. The top 10 upregulated and downregulated pathways

are shown. **e** mRNA levels of *Ifng* in the infected skin on day 4 p.i were measured by RT-qPCR. Data were normalized to the housekeeping gene *Ube2d2*. Pooled data from 2 independent experiments are shown and mean values ± SEM are given; PBS: n = 2/genotype; *C. albicans*: n = 12 (*WT*), n = 7 (*Tyk2⁻/⁻*) and n = 9 (*Tyk2^K923E*); n.d not detectable; n: biological replicates. **f** IFNγ in the skin on day 2 p.i was measured using a Luminex assay. Pooled data from 2 independent experiments are shown. Mean values ± SEM are given; PBS: n = 2/genotype; *C. albicans*: n = 8 (WT) and n = 9 (*Tyk2⁻/⁻, Tyk2^K923E*); n.d, not detectable; n: biological replicates. Statistical analysis was conducted using One-way ANOVA followed by Tukey's multiple comparison test and statistical significance is only given for the comparison between the genotypes (**e, f**). Source data are provided as a Source Data file.

(ii) phagocytosis, such as *Cd302*, *Icam5* and *Tgm2*, and (iii) neutrophil maturation, such as *Klf6* and *Runx1* (Supplementary Data).

To validate our RNA-sequencing data, we analyzed the expression of well-known ISGs in skin biopsies via RT-qPCR and found strongly impaired expression of *Stat1*, *Oas1a*, *Rsad2* and *Sp100*, in infected *Tyk2*$^{-/-}$ and *Tyk2*$^{K923E}$ mice when compared to *WT* mice (Supplementary Fig. S4b).

Collectively, these results indicate that intradermal infection with *C. albicans* induces an IFN response in skin-infiltrating neutrophils that requires active TYK2 signaling. The RNA-sequencing results furthermore indicate that TYK2 signaling does not affect the expression of genes associated with neutrophil maturation, migration, and phagocytic potency.

## IFNγ facilitates *C. albicans* dissemination to kidneys following intradermal infection

ISGs can be induced by all types of IFNs, albeit with gene-specific quantitative differences[30–32]. Type I IFNs (IFNα/β) and type II IFNs (IFNγ) act on most if not all cell types, whereas type III IFNs (IFNλ) primarily act on epithelial cells. Previous studies have shown that *C. albicans* infection into the deep skin induces early production of IFNγ[27,33]. We found strong TYK2-dependent upregulation of IFNγ at the transcriptional and protein level in the skin at day 4 p.i (Fig. 4e, f), whereas the upregulation of IL-1β, IL-23, and IL-10 occurred independently of TYK2 (Supplementary Fig. S4c, d).

To test the impact of IFNγ on *C. albicans* infection in our experimental model, we employed mice that lack a functional IFNγ receptor (*Ifngr1*$^{-/-}$ mice) (Fig. 5a). *Ifngr1*$^{-/-}$ mice had a comparable fungal burden in the skin at day 4 p.i (Fig. 5b, Supplementary Fig. S5a) but displayed a significantly lower fungal burden in the kidneys when compared to *WT* mice (Fig. 5c, Supplementary Fig. S5b), indicating that IFNγ promotes fungal dissemination to distal organs but does not affect local fungal growth. Similar to our observations in *Tyk2*$^{-/-}$ and *Tyk2*$^{K923E}$ mice, *Ifngr1*$^{-/-}$ mice showed less pronounced changes in the white blood cell composition than *WT* mice (Fig. 5d), a better containment of *C. albicans* at the infection site, as indicated by GMS staining of skin sections (Fig. 5e), and the formation of a Ki-67-negative layer around the pathogen (Fig. 5f).

Like *Tyk2*$^{-/-}$ mice, *Ifngr1*$^{-/-}$ mice did not differ from *WT* mice with respect to the number of infiltrating neutrophils and the abundance of monocytes and macrophages in the skin after *C. albicans* infection (Fig. 5g, Supplementary Fig. S5c).

Overall, these results show that IFNγ promotes *C. albicans* invasion into deep skin layers after intradermal infection and facilitates systemic fungal dissemination to kidneys.

## *Rag2*$^{-/-}$ mice lack IFNγ upregulation in the skin and show profoundly reduced fungal dissemination to kidneys

To test whether TYK2 in innate or adaptive immune cells is required for IFNγ production and *C. albicans* dissemination to kidneys, we crossed TYK2-mutant mice with *Rag2*$^{-/-}$ mice, the latter lacking mature T cells, B cells, and NKT cells, but still containing NK cells[34], and infected them with *C. albicans* (Supplementary Fig. S6a). *Rag2*$^{-/-}$*Tyk2*$^{-/-}$ and *Rag2*$^{-/-}$*Tyk2*$^{K923E}$ mice showed a comparable fungal burden in the skin as *Rag2*$^{-/-}$ control animals on day 4 p.i, as well as similar infection-associated changes in the white blood cell composition (Supplementary Fig. S6b, d). Surprisingly, we barely observed fungal invasion into deeper skin layers or dissemination to kidneys, irrespective of the presence or absence of catalytically active TYK2 (Supplementary Figs S6e, c). Additionally, IFNγ production was undetectable in mice of all three genotypes. To further corroborate the latter findings, we directly compared *WT* and *Rag2*$^{-/-}$ mice (Fig. 6a). We observed that contrary to *WT* mice, *Rag2*$^{-/-}$ animals fail to upregulate IFNγ on day 4 p.i (Fig. 6b). Moreover, *Rag2*$^{-/-}$ mice displayed slightly higher fungal load in the skin (Fig. 6c, Supplementary Fig. S6f) and, most importantly,

showed a profoundly reduced fungal burden in the kidneys when compared to *WT* mice (Fig. 6d, Supplementary Fig. S6g). In line with the reduced dissemination, we also observed less *C. albicans* invasion into deeper skin in *Rag2*$^{-/-}$ than in *WT* mice (Fig. 6e). We did not observe differences in the abundance of neutrophils, monocytes, and macrophages between *Rag2*$^{-/-}$ and *WT* mice (Fig. 6f, Supplementary Fig. S6h).

Taken together, these results indicate that the disease-promoting functions of TYK2 depend on the presence of T cells and/or B cells and that these cells contribute to the control of *C. albicans* growth in the skin but facilitate fungal dissemination to kidneys. Moreover, the data suggest that T cells or NKT cells, but not NK cells, are a major source of IFNγ in response to intradermal *C. albicans* infection.

## Ablation of TYK2 in αβ T cells and NKT cells is not sufficient to protect from *C. albicans* dissemination to kidneys

To further investigate which T cell subtype produces IFNγ upon intradermal *C. albicans* infection, we employed conditional *Tyk2* knockout mice (*Tyk2*$^{fl/fl}$) crossed with *CD4-Cre* mice, to ablate *Tyk2* in αβ T cells and NKT cells, whereas γδ T cells retain *Tyk2* expression (*Tyk2*$^{ΔT}$ mice)[35–37] (Fig. 7a). Fungal loads in the skin and kidneys were comparable between *Tyk2*$^{fl/fl}$ and *Tyk2*$^{ΔT}$ mice (Fig. 7b, c, Supplementary Fig. S7a, b). In line with these data, we did not observe differences between *Tyk2*$^{fl/fl}$ and *Tyk2*$^{ΔT}$ mice in the frequency of blood granulocytes and lymphocytes (Fig. 7d), *Ifng* mRNA and IFNγ protein levels in the skin (Fig. 7e, f), monocyte/neutrophil and macrophage infiltration (Fig. 7g, j, Supplementary Fig. S7c), the distribution of Ki-67$^+$ cells around the injection site (Fig. 7h) or fungal invasion onto deeper skin layers (Fig. 7i).

Deletion of *Tyk2* in αβ T cells did not alter the abundance of αβ T cells, γδ T, and NK cells in the skin and the skin-draining lymph nodes (Supplementary Fig. S7d–g).

Collectively, these results demonstrate that deletion of *Tyk2* specifically in αβ T cells and NKT cells does not (i) influence local fungal growth and dissemination to kidneys and (ii) impair IFNγ production in the skin.

## γδ T cells produce IFNγ in response to *C. albicans* in a TYK2-dependent manner

γδ T cells are innate-like lymphocytes with high prevalence in mouse skin and an important role in the maintenance of tissue homeostasis and protection against infections[9]. In the context of cutaneous candidiasis, dermal γδ T cells have been reported to increase host resistance through the production of IL-17A[26,38]. In a *Staphylococcus aureus* skin infection model, γδ T cells produce IFNγ, which protects against subsequent infections[39]. To investigate whether murine γδ T cells are capable of producing IFNγ in response to *C. albicans*, we stimulated *WT* and *Tyk2*$^{-/-}$ skin cells with heat-killed *C. albicans* and analyzed IFNγ production by γδ T cells using flow cytometry (Fig. 8a, Supplementary Fig. S8a). IL-12 was used as a positive control because it is a potent inducer of IFNγ and requires TYK2 for signaling[40]. *C. albicans* stimulation induced IFNγ production by *WT* γδ T cells to similar levels as IL-12 (Fig. 8b–d). In contrast, *Tyk2*$^{-/-}$ γδ T cells showed strongly impaired IFNγ production in response to both stimuli (Fig. 8b–d). Similar results were observed with splenic γδ T cells (Supplementary Fig. 8b–e). These findings demonstrate that γδ T cells from both skin and spleen produce IFNγ in response to *C. albicans* infections in a TYK2-dependent manner.

## Discussion

Fungal infections are considered hidden killers of mankind, as they claim an extreme number of lives every year[41]. *C. albicans* is the most common fungal pathogen in humans, causing diseases ranging from local (muco)cutaneous to life-threatening systemic infections[42,43]. In this study, we report that TYK2-mediated immune signaling increases the sensitivity to *C. albicans* skin infections, which is in sharp contrast to the protective role of TYK2 in the host defense against bacterial and

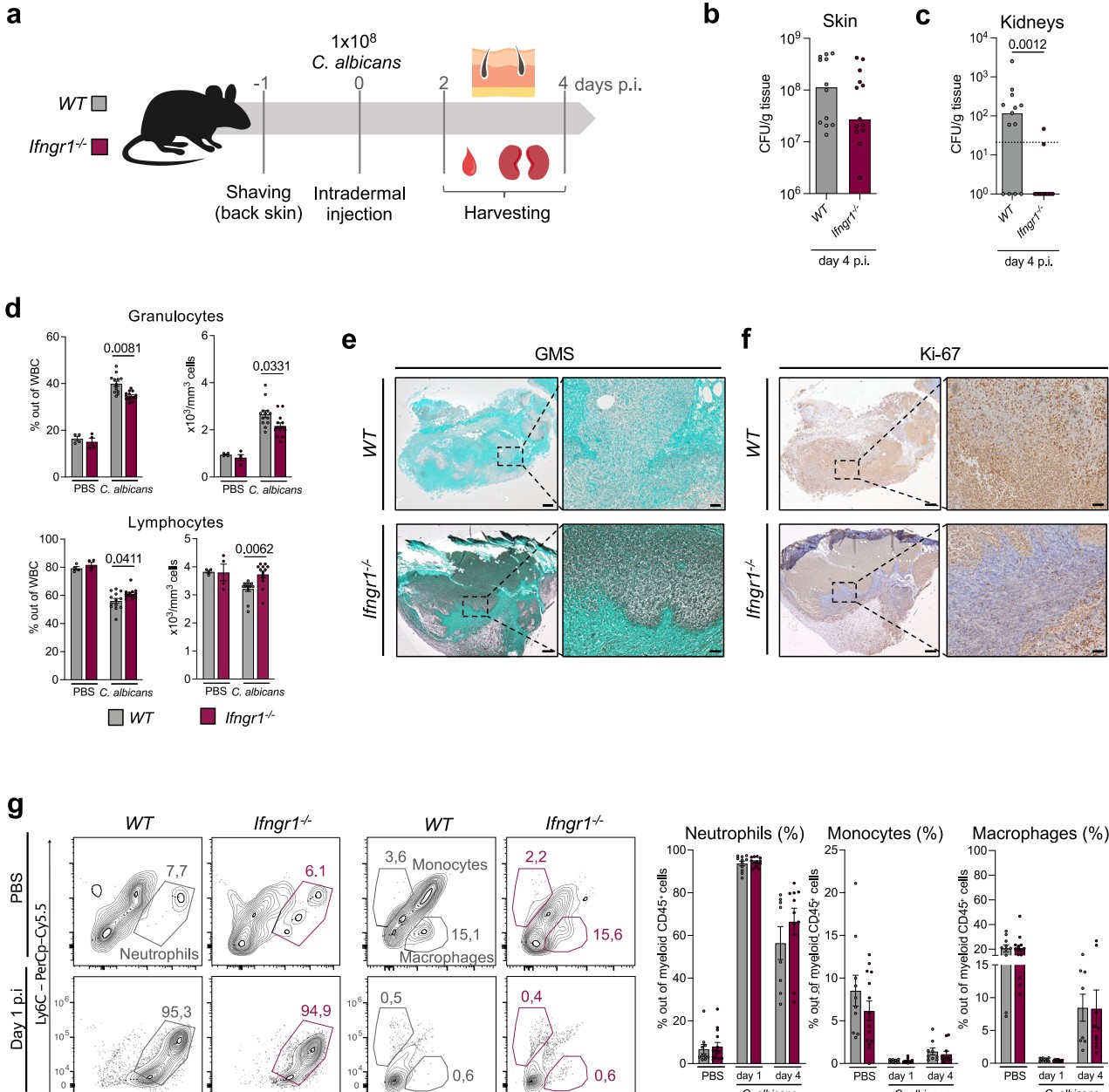

**Fig. 5 | *Ifngr1*⁻/⁻ mice are less sensitive to cutaneous candidiasis than *WT* mice and grossly phenocopy TYK2-mutants. a** *WT* and *Ifngr1*⁻/⁻ were infected as described in the legend to Fig. 1. Fungal load in the skin (**b**) and in the kidneys (**c**) was measured on day 4 p.i. Pooled data from 2 independent experiments are shown. Mean values ± SEM are given; skin: n = 12 (*WT*) and n = 13 (*Ifngr1*⁻/⁻); kidneys: n = 13/genotype; n: biological replicates. The dotted line indicates the assay detection limit (**b**, **c**). **d** Blood cell composition on day 4 p.i. was determined with a VetABC analyzer. The percentage of granulocytes and lymphocytes out of total white blood cells (WBC) and the total numbers of these populations in the blood are shown. Pooled data from 2 independent experiments are shown. Mean values ± SEM are given; PBS: n = 4/genotype, *C. albicans:* n = 13/genotype; n: biological replicates. Representative pictures of the infected skin on day 4 p.i. Sections were stained with GMS (**e**) or an anti-Ki-67 antibody (**f**). Scale bar: 200 μm (left),

50 μm (right). Data are representative of sections from 8 (*WT*) and 7 (*Ifngr1*⁻/⁻) mice. **g** The percentages of skin-infiltrating neutrophils (gated as CD45⁺CD11b⁺Ly6C⁺Ly6G⁺ cells), monocytes (gated as CD45⁺CD11b⁺Ly6G⁻Ly6C^high F4/80⁻ cells) and macrophages (gated as CD45⁺CD11b⁺Ly6G⁻F4/80⁺ cells) out of CD45⁺CD11b⁺ cells were determined by flow cytometry analysis on day 1 and 4 p.i. Representative contour plots and pooled data from 2 independent experiments for each time-point are shown. Mean values ± SEM are given; PBS: n = 10 (*WT*) and n = 14 (*Ifngr1*⁻/⁻); day 1: n = 11 (*WT*) and n = 13 (*Ifngr1*⁻/⁻); day 4: n = 8 (*WT*) and n = 10 (*Ifngr1*⁻/⁻) for neutrophils and macrophages and n = 9 (*WT*) and n = 11 (*Ifngr1*⁻/⁻) for monocytes; n: biological replicates. Statistical analysis was conducted using a two-tailed Mann Whitney test (**b**, **c**) and One-way ANOVA followed by Tukey's multiple comparison test (**d**, **g**). Statistical significance is only given for the comparison between the genotypes (**b**–**d**, **g**). Source data are provided as a Source Data file.

viral infections[15,16]. We show that the absence of TYK2 or its catalytic activity decreases dissemination of *C. albicans* to kidneys in response to intradermal infection and facilitates wound healing and recovery from the infection. Furthermore, we show that the absence of

functional IFNγ signaling prevents *C. albicans* invasion into deeper skin tissues as well as systemic dissemination to kidneys and that innate-like T cells are the major source of IFNγ in response to intradermal *C. albicans* infection.

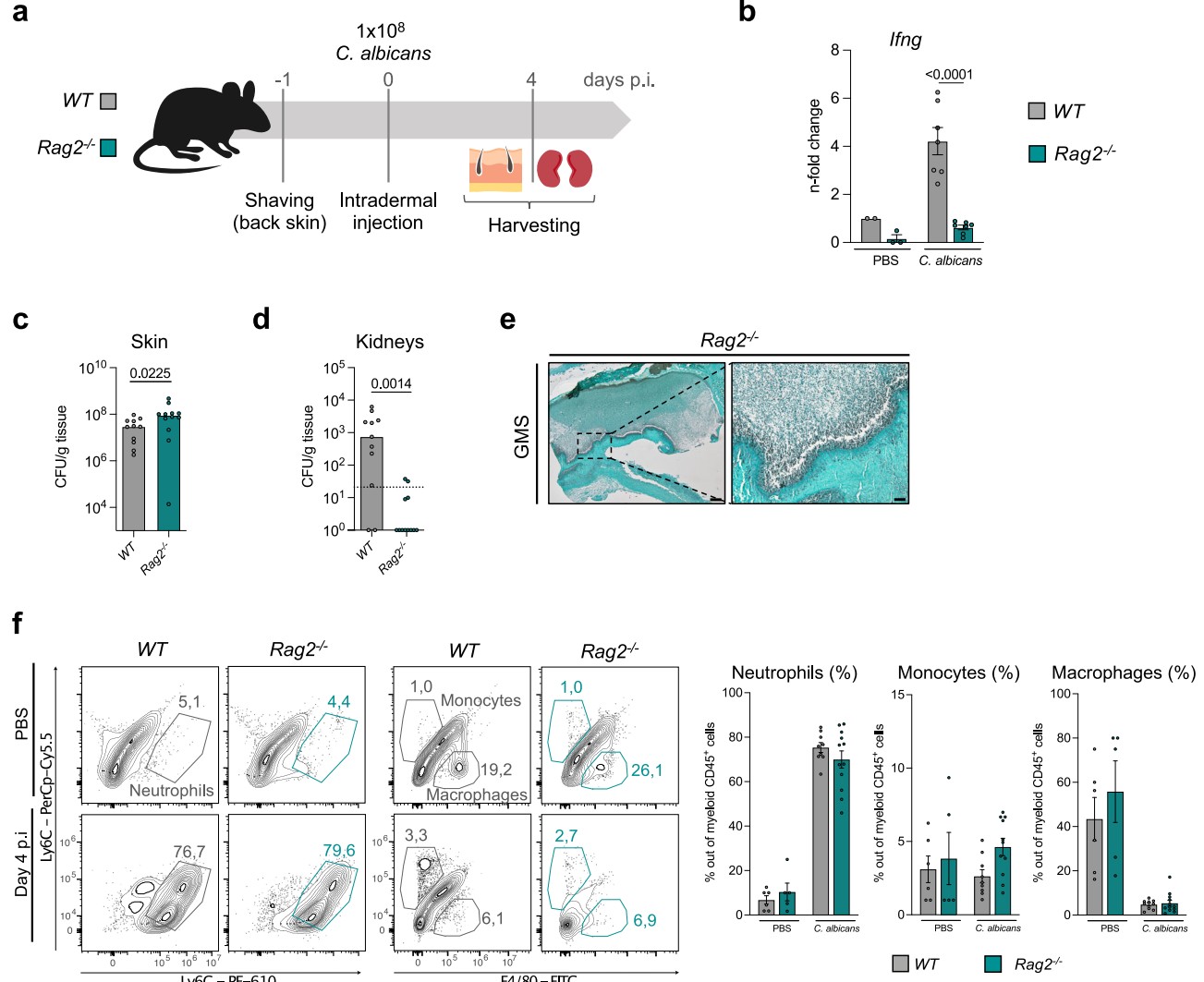

**Fig. 6 | RAG2 deficiency leads to diminished *C. albicans* tissue dissemination and impaired IFNγ production. a** *WT* and *Rag2⁻/⁻* mice were infected as described in the legend to Fig. 1. **b** mRNA levels of *Ifng* in the skin of *WT* and *Rag2⁻/⁻* mice on day 4 p.i were measured by RT-qPCR. Data were normalized to the housekeeping gene *Ube2d2* and fold changes were calculated relative to the uninfected *WT* controls. Pooled data from 2 independent experiments are shown. Mean values ± SEM are given; PBS: n = 2 (*WT*) and n = 3 (*Rag2⁻/⁻*); *C. albicans*: n = 7/genotype; n: biological replicates. Statistical significance is only given for the comparison between the genotypes). Fungal load in the skin (**c**) and in the kidneys (**d**) was determined on day 4 p.i by CFU assay. Pooled data from 2 independent experiments are shown. Median values are given; n = 11 (*WT*) and n = 12 (*Rag2⁻/⁻*); n: biological replicates. The dotted line indicates the assay detection limit. **e** Representative pictures of the infected skin on day 4 p.i. A GMS staining of the skin sections is shown. Data are representative of sections from 4 mice per genotype. Scale bar: 200 μm (left), 50 μm (right). **f** The percentages of skin-infiltrating neutrophils (gated as CD45⁺CD11b⁺Ly6C⁺Ly6G⁺ cells), monocytes (gated as CD45⁺CD11b⁺Ly6G⁻Ly6C^high F4/80⁻ cells) and macrophages (gated as CD45⁺CD11b⁺Ly6G⁻F4/80⁺ cells) out of CD45⁺CD11b⁺ cells were determined by flow cytometry analysis on day 4 p.i. Representative contour plots and pooled data from 2 independent experiments are shown. Mean values ± SEM are given; PBS: n = 6 (*WT*) and n = 5 (*Rag2⁻/⁻*); *C. albicans*: n = 9 (*WT*) and n = 12 (*Rag2⁻/⁻*) for neutrophils and macrophages; n = 9 (*WT*) and n = 11 (*Rag2⁻/⁻*) for monocytes; n: biological replicates. Statistical analysis was conducted using a two-tailed Mann Whitney test (**c**, **d**) and One-way ANOVA followed by Tukey's multiple comparison test (**b**, **f**). Statistical significance is only given for the comparison between the genotypes (**b**–**d**, **f**). Source data are provided as a Source Data file.

Neutrophils are among the first line of defense against mucosal and systemic *C. albicans* infections[6,44,45]. In cutaneous candidiasis, these cells are recruited to the skin within the first hours of infection and are associated with better fungal control[27,33]. We show that the absence of TYK2 does not impact the initial recruitment of neutrophils to the infection site but results in the accumulation of necrotic neutrophils organized in a barrier-like structure that surrounds the fungal cells. A similar phenotype was observed in mice lacking a functional IFNγ receptor. TYK2 is not directly involved in IFNγ signaling but it promotes IFNγ production through its well-established role in the IL-12 and IL-23 signaling cascades[14,17]. Therefore, one may infer that TYK2 regulates neutrophils through its function upstream of IFNγ in our experimental model. In line, we found impaired local production of IFNγ and impaired ISG expression in skin-infiltrating neutrophils from *Tyk2⁻/⁻* and *Tyk2^K923E* mice. Previous work has shown that IFNγ increases the lifespan of human neutrophils in vitro[46–48], raising the exciting possibility that the necrotic neutrophil layer around *C. albicans* results from accelerated neutrophil death in the absence of TYK2 or IFNγ signaling. Surprisingly, the presence of the necrotic cell layer around the wound in both *Tyk2*- and *Ifngr1*-deficient mice correlates with better fungal containment at the infection site and impaired fungal dissemination to kidneys. Moreover, we show that neutrophil-depletion abolishes the difference in fungal dissemination between *WT* and *Tyk2⁻/⁻* mice, supporting the conclusion that neutrophils crucially contribute to the better containment of *C. albicans* at the infection site in *Tyk2⁻/⁻*

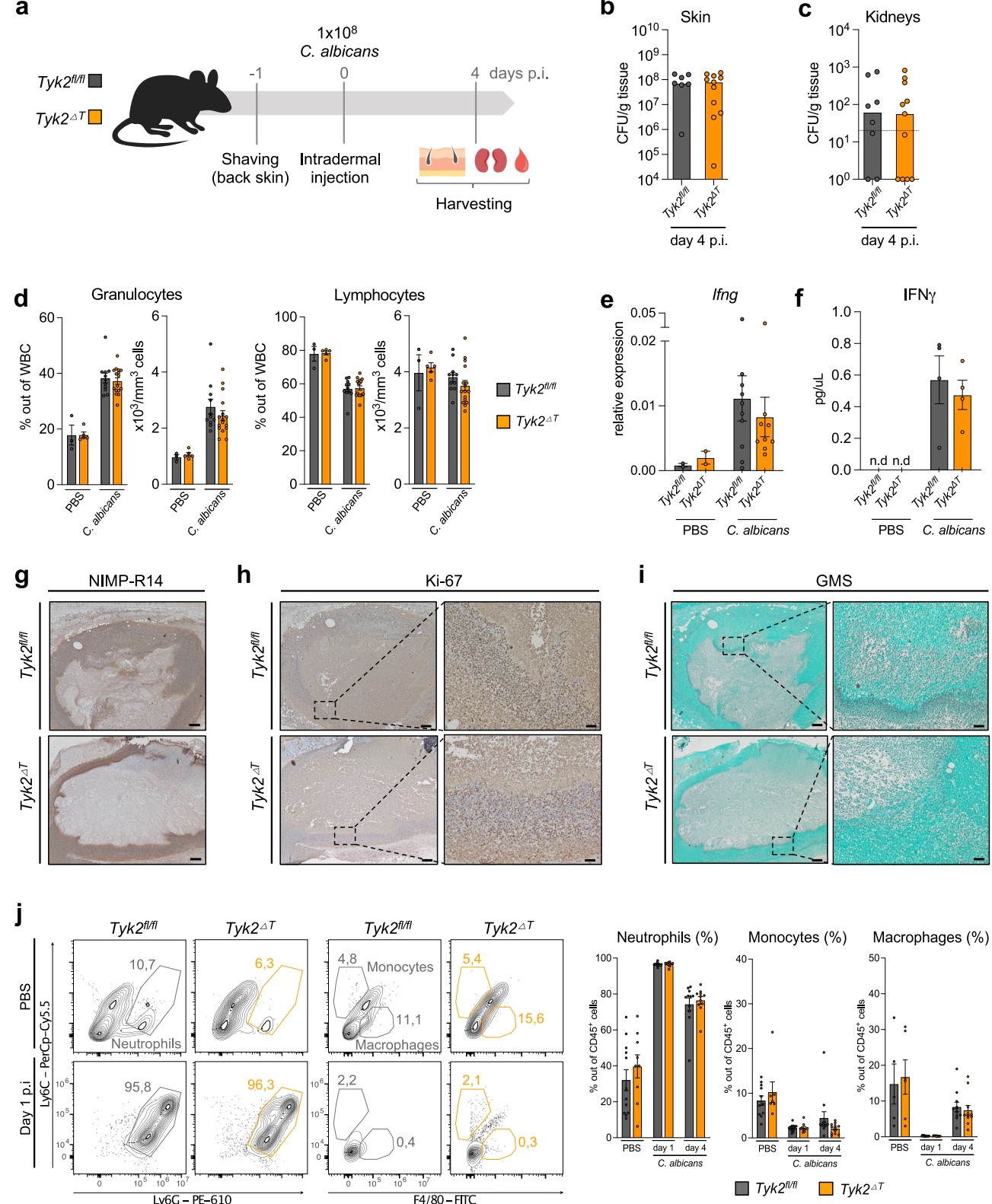

mice. Hence, it is tantalizing to speculate that even necrotic or inviable neutrophils can carry a mechanical antifungal effector function by forming barrier layers.

It remains unclear if the dead neutrophils per se act as a barrier that inhibits fungal migration into deeper skin tissues and vasculature-driven dissemination to peripheral organs. Additional TYK/IFNγ-dependent mechanisms could be involved. For instance, we can not

exclude a potential role of neutrophil extracellular traps (NETs) in the formation of such neutrophil "road blocks", as NET detection in situ is very challenging[49,50]. Moreover, TYK2/IFNγ signaling may increase the clearance of necrotic neutrophils by (recruited) macrophages within infected tissues.

Our findings complement the previously described disease-promoting function of IFNγ in mucosal candidiasis. Exacerbated IFNγ

**Fig. 7 | Deletion of TYK2 in αβ T cells and NKT cells is not sufficient to protect mice against candidiasis. a** $Tyk2^{\Delta T}$ and $Tyk2^{fl/fl}$ mice were infected as described in the legend to Fig. 1. Fungal load in the skin (**b**) and in the kidneys (**c**) was measured on day 4 p.i. Pooled data from 2 independent experiments are shown. Median values are given; skin: n = 7 ($Tyk2^{fl/fl}$) and n = 11 ($Tyk2^{\Delta T}$); kidneys: n = 8 ($Tyk2^{fl/fl}$) and n = 11 ($Tyk2^{\Delta T}$); n: biological replicates. The dotted line indicates the assay detection limit. **d** Blood cell composition on day 4 p.i was determined with a Vet ABC analyzer. The percentage of granulocytes and lymphocytes out of total white blood cells (WBC) and total numbers of these populations in the blood are shown. Pooled data from 2 independent experiments are shown. PBS: n = 3 ($Tyk2^{fl/fl}$) and n = 5 ($Tyk2^{\Delta T}$); *C. albicans*: n = 11 ($Tyk2^{fl/fl}$) and n = 16 ($Tyk2^{\Delta T}$); n: biological replicates. **e** *Ifng* mRNA levels in the skin on day 4 p.i were measured by RT-qPCR. Pooled data from 2 independent experiments are shown. PBS: n = 2/genotype; *C. albicans*: n = 10 ($Tyk2^{fl/fl}$) and n = 9 ($Tyk2^{\Delta T}$); n: biological replicates. **f** IFNγ protein levels on day 4 p.i were measured using a Luminex assay. Pooled data from 2 independent experiments are shown. PBS: n = 2/genotype; *C. albicans*: n = 4/genotype; n.d, not detectable; n: biological replicates. Representative pictures of the infected skin on day 4 p.i. Sections were stained with anti-NIMP-

R14 (**g**) or anti-Ki-67 (**h**) antibodies or with GMS staining (**i**). Scale bar: 200 μm (left), 50 μm (right). Data are representative of sections from 4 ($Tyk2^{fl/fl}$) and 3 ($Tyk2^{\Delta T}$) mice (**g–i**). **j** The percentages of skin-infiltrating neutrophils (gated as CD45⁺CD11b⁺Ly6C⁺Ly6G⁺ cells), monocytes (gated as CD45⁺CD11b⁺Ly6G⁻Ly6C^{high}F4/80⁻ cells) and macrophages (gated as CD45⁺CD11b⁺Ly6G F4/80⁺ cells) out of CD45⁺CD11b⁺ cells were determined by flow cytometry analysis on day 1 and 4 p.i. Representative contour plots and pooled data from 2 independent experiments for each time-point are shown. PBS - n = 12 ($Tyk2^{fl/fl}$), n = 9 ($Tyk2^{\Delta T}$) for Neutrophils; n = 11 ($Tyk2^{fl/fl}$), n = 7 ($Tyk2^{\Delta T}$) for Monocytes; n = 5 ($Tyk2^{fl/fl}$), n = 6 ($Tyk2^{\Delta T}$) for Macrophages; *C. albicans* day 1 - n = 14 ($Tyk2^{fl/fl}$), n = 10 ($Tyk2^{\Delta T}$); *C. albicans* day 4 - n = 11/genotype; n: biological replicates; Mean values ± SEM are given (**d–f**, **j**). Statistical analysis was conducted using a two-tailed Mann Whitney test (**b**, **c**) and One-way ANOVA followed by Tukey's multiple comparison test (**d–f**). Statistical analysis between genotypes has not been done for PBS controls in (**e**, **f**), as the sample size is <3. Statistical significance is only given for the comparison between the genotypes (**b–f**, **j**). Source data are provided as a Source Data file.

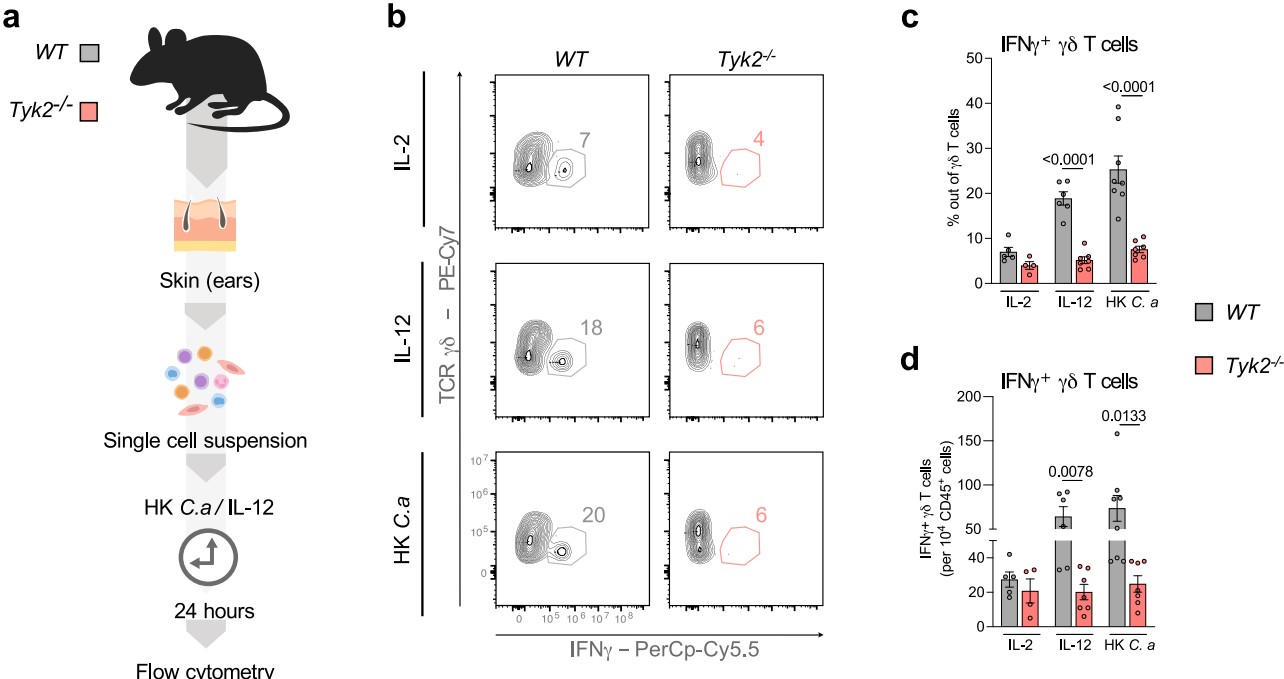

**Fig. 8 | γδ T cells produce IFNγ upon *C. albicans* stimulation. a** The skin of *WT* and $Tyk2^{-/-}$ mice was collected, digested and single cell suspensions were prepared. 2 × 10⁶ cells were incubated for 24 h with heat-killed *C. albicans* (HK *C.a*), IL-12 or IL-2 and stained for flow cytometry analysis. The percentage of IFNγ⁺ cells out of γδ T cells (**b**, **c**) and the total numbers of these cells (**d**) are shown. Representative contour plots (**b**) and pooled data from 2 independent experiments are shown

(**c**, **d**). Mean values ± SEM are given; IL-2: n = 5 (*WT*) and n = 4 ($Tyk2^{-/-}$); IL-12: n = 6 (*WT*) and n = 7 ($Tyk2^{-/-}$); HK *C.a*: n = 8 (*WT*) and n = 7 ($Tyk2^{-/-}$); n: biological replicates. Statistical analysis was conducted using One-way ANOVA followed by Tukey's multiple comparison test and statistical significance is only given for the comparison between the genotypes (**c**, **d**). Source data are provided as a Source Data file.

production or STAT1 activity are associated with an increased sensitivity to chronic mucosal candidiasis (CMC). STAT1 gain-of-function (GOF) mutations are among the most common genetic associations with CMC and are characterized by an impaired IL-17-dependent mucosal defense against *C. albicans*[51–53]. A recent study discovered that IFNγ increases the sensitivity to oropharyngeal candidiasis by promoting epithelial pathology in the context of unimpaired IL-17 signaling[54]. CMC patients rarely develop invasive candidiasis[55]. Our findings would predict that these patients have an increased risk of disseminated candidiasis if the pathogen reaches deeper skin layers. It would be interesting to test this hypothesis, for example by using mice expressing STAT1-GOF variants[56,57].

Furthermore, in this work we present compelling genetic evidence that innate-like T cells are the primary source of tissue IFNγ in response to intradermal *C. albicans* infection. $Rag2^{-/-}$ mice, which lack mature B and T cells, fail to induce IFNγ in the skin, ruling out NK cells as major source for IFNγ in our experimental model. This contrasts with an early report suggesting unimpaired IFNγ production in $Rag2^{-/-}$ mice and NK cells as the primary IFNγ source within the first hours after infection with *C. albicans* into the deep dermis[33]. Given that B cells are best-known for producing pathogen-specific antibodies and do not produce IFNγ in response to fungal pathogens[58,59], our data suggest that IFNγ is produced by T cells in our experimental model. $Tyk2^{\Delta T}$ mice, which lack TYK2 specifically in both αβ T cells and NKT cells but

not in γδ T cells, still produce IFNγ in response to *C. albicans* infection, arguing against a contribution of αβ T cells or NKT cells to the TYK2-driven IFNγ production. In the immune response to local fungal infections, the role of γδ T cells has been mainly associated with their IL-17-producing capacity[8,10]. During cutaneous candidiasis, IL-17-producing dermal γδ T cells have been shown as essential for *C. albicans* growth control[8,27,38]. Remarkably, we show here that skin-derived γδ T cells are capable of producing IFNγ in response to *C. albicans* in vitro and that this depends on the presence of enzymatically active TYK2.

Another key finding of our study is that *Tyk2^{-/-}* and *Tyk2^{K923E}* mice show a similar reduction in *C. albicans* invasion into the deep skin and dissemination to kidneys when compared to wild-type controls, suggesting that TYK2 kinase inhibition might be an option to treat cutaneous candidiasis. Further studies are needed to investigate the potential beneficial effects of TYK2 inhibitors on the immune defense against *C. albicans* skin infections. Our results would also predict that the use of TYK2 inhibitors for the treatment of psoriasis or other autoimmune and inflammatory diseases[20] would not increase the risk of *C. albicans* skin infections.

The major limitation of our study is that it remains unclear how our results translate to human patients. The intradermal infection model used in our study mimics infection of the deeply wounded skin, which may occur upon barrier dysfunction, physical skin injury (e.g., through introduction of medical devices), or burns, and only causes fungal dissemination with high infectious doses. Disseminated candidiasis originating from skin is a rare but serious condition. It can occur in patients with both primary and acquired immunodeficiency, such as patients with *CARD9* mutation, chronic granulomatous disease (CGD), or chemotherapy-induced neutropenia[55,60–62]. The steady rise in the use of various antibiotics may also increase the risk for disseminated skin infections even in immunocompetent patients in conjunction with burns or extensive invasive procedures. Whether or not these patients would benefit from TYK2 inhibition remains to be investigated. However, it must be taken into consideration that TYK2 inhibition may increase the sensitivity to other microbial infections and impair the immune response against cancer cells, albeit to a lesser extent than JAK1/JAK2 inhibitors[20,63]. Moreover, it remains to be investigated how TYK2 deficiency affects the antifungal response when *C. albicans* is directly delivered to the blood stream. Another limitation of our study is the lack of direct evidence for γδ T cells being a major source of IFNγ in our experimental model.

Taken together, our study unravels a novel and as yet unrecognized detrimental role of TYK2 and IFNγ signaling in the immune defense against invasive cutaneous candidiasis and paves the way for exploring TYK2 inhibitors as potential treatment option.

## Methods

### Ethics statement
All gene-targeted and wild-type mice used in this study were bred at the University of Veterinary Medicine Vienna under specific-pathogen-free conditions according to Federation of European Laboratory Animal Science Associations (FELASA) guidelines. All animal experiments were approved by the ethics and animal welfare committee of the University of Veterinary Medicine Vienna and the Austrian Federal Ministry of Science and Research according to §§ 26ff. of Animal Experiments Act, Tierversuchsgesetz 2012−TVG 2012 (BMWFW-68.205/0032- WF/II/3b/2014 and BMBWF-68.205/0173-V/3b/2019) and conform to the guidelines of FELASA and ARRIVE (Animal Research: Reporting of In Vivo Experiments).

### Mice
Wild-type (*WT*, C57BL/6 N), *Tyk2^{-/-}*, *Tyk2^{K923E}*, *Tyk2^{fl/fl}* and *Ifngr1^{-/-}* mice (all on C57BL/6 background) were previously described[25,64–66]. To generate mice that lack TYK2 in αβ T cells (*Tyk2^{ΔT}*), *Tyk2^{fl/fl}* mice were

crossed to *CD4-cre* mice[37] (*B6.Cg-Tg(Cd4-cre)^{1Cwi}*, kindly provided by Wilfried Ellmeier, Medical University of Vienna). *Rag2^{-/-}* (*B6.129S6-Rag2^{tm1Fw}*) mice[34] were kindly provided by Veronika Sexl, University of Veterinary Medicine Vienna. To generate mice that lack both RAG2 and TYK2 or express enzymatically inactive TYK2, *Rag2^{-/-}* mice were crossed with *Tyk2^{-/-}* and *Tyk2^{K923E}* mice, respectively. Age- and sex-matched (8–12 weeks) mice were used. All mice were housed under controlled 12 h light/dark cycles (light period 6:00 am–6:00 pm) with room temperatures of 20−22 °C and relative humidity of 55 ± 10%.

### *C. albicans* culture and in vivo infection
The *C. albicans* strain SC5314[67] was grown at 30 °C in YPD medium (10 g/L yeast extract (YEA03; Formedium, Norfolk, England), 20 g/L peptone (PEP03; Formedium), 2% (w/v) D-Glucose (141341.0914; RanReac Appli-Chem, Chicago, USA)). Growth was monitored by optical density (OD$_{600}$) measurement. Once cultures reached the logarithmic growth phase (OD$_{600}$ = 1, concentration $1 \times 10^7$ CFU/mL), cells were centrifuged at $1780 \times g$, 21 °C, for 5 min, washed two times with PBS, and resuspended in PBS to obtain the desired infection concentration. To confirm the infection dose, an aliquot was plated on YPD-agar plates. On the day before the infection, mice were anesthetized by intraperitoneal injection of Ketamine/Xylazine (80 mg/kg BW Ketamine (Ketamidor; VetViva, Richter GmbH, Wels, Austria), 4 mg/kg BW Xylazine (Sedaxylan; Eurovet Animal Health, Bladeln Netherlands) for shaving the back skin. On the infection day, mice were anesthetized in the same way and intradermally (i.d) infected with $1 \times 10^8$ CFU *C. albicans* in 50 μL (2 independent 25 μL injection sites) or mock infected with PBS. For preparation of heat-killed *C. albicans*, fungal suspensions were incubated at 65 °C for 15 min with continuous shaking. Proper fungal killing was evaluated by plating the suspension on YPD-agar plates.

### Colony-forming unit (CFU) assay
To determine fungal burden, skin biopsies with a 4 mm diameter and kidneys were harvested in cold PBS at different timepoints post-infection, weighed and homogenized using a pellet pestle. Serial dilutions were plated in duplicates on YPD-agar plates, and colonies were counted after 48 h growth at 37 °C.

### Immunohistochemistry
Skin biopsies were fixed in 4% formaldehyde solution for 24 h at room temperature, washed with 70% ethanol, dried and embedded in paraffin. For immunohistochemical staining, samples were cut into 3 μm sections, deparaffinized, and rehydrated. Endogenous peroxidase activity was blocked and antigen retrieval was performed. Sections were then stained with primary and secondary antibodies according to manufacturer's protocol. NIMP-R14 (ab2557; Abcam, Cambridge, United Kingdom), anti-Ki-67 (14-5698-82; eBioscience/Thermofisher) and anti-MPO (AF3667-SP; Bio-techne, Minneapolis, USA) antibodies were used. As a secondary antibody, goat anti-rat IgG:Biotin (STAR131B; Bio-Rad) or swine anti-rabbit (E0353; DAKO-Agilent, Santa Clara, California, USA) was used. Stainings were visualized with the Vectastain ABC kit (PK-6100; Vector labs, San Francisco, California, USA) and DAB substrate (SK-4100; Vector labs) and sections were counterstained with haematoxylin (GHS316; Sigma-Aldrich, St. Louis, Missouri, United States). To stain *C. albicans*, a Gomori Methenamine-Silver Nitrate (GMS) Stain Kit (LGA125; Scytec, Utah, United States) was used according to manufacturer's instructions. For each timepoint and IHC/GMS staining, one skin section per mouse (n = 2, PBS; n = 6-9, *C. albicans*) was stained and examined under the microscope. Pictures were taken from all sections and representative images were selected for the figures of the manuscript.

### RNA isolation and reverse transcription quantitative PCR
Skin biopsies were homogenized in RLT buffer from RNeasy Fibrous Tissue Mini Kit (74704; Qiagen, Hilden, Germany) using 1600 MiniG

tissue homogenizer (3 cycles of 1500 RPM, 60 s). RNA was isolated according to the manufacturer's instructions. RNA purity was determined by Nanodrop and agarose gel electrophoresis. 1 µg of RNA was reverse transcribed using the iScript cDNA Synthesis Kit (1708891; Bio-Rad, Hercules, California, USA). Quantitative PCR was performed with a CFX96 Touch Real-Time PCR Detection System (Bio-Rad) using *Ube2d2* as housekeeping gene. The following primers were used: *Ube2d2*-forward 5′-AGG TCC TGT TGG AGA TGA TAT GTT-3′, *Ube2d2*-reverse 5′-TTG GGA AAT GAA TTG TCA AGA AA-3′, *Ube2d2*-probe 5′-CCA AAT GAC AGC CCC TAT CAG GGT GG-3′, *Ifng*-forward 5′-TGA GTA TTG CCA AGT TTG AGG TCA-3′, *Ifng*-reverse 5′-CGG CAA CAG CTG GTG GAC-3′, *Ifng*-probe 5′-CCA GCG CCA AGC ATT CAA TGA GCT-3′. QuantiTect Primer Assays Mm_Il23a_2_SG (QT01663613) and Mm_Il1b_2_SG (QT01048355) (Qiagen, Hilden, Germany) were reconstituted according to manufacturer's instructions. HotFIREPol DNA Polymerase (Solis BioDyne, Tartu, Estonia), EvaGreen (Biotium Inc., Fremont, California, USA) and dNTP Set (ThermoFisher, Waltham, Massachusetts, USA) were used according to manufacturer's instructions.

## Cytokine measurements (Luminex assays)

Skin samples were homogenized in Schindler's buffer[25]. The levels of IFNγ, IL-1β, and IL-23 in skin homogenates were determined using ProcartaPlex Immunoassays (ThermoFisher) according to manufacturer's protocol. ProcartaPlex Mouse Basic Kit (EPX010-20440-901) was used together with ProcartaPlex™ Simplex Kits for IFNγ (EPX01A-20606-901), IL-1β (EPX01A-26002-901), and IL-23 (EPX01A-26017-901), and analyzed on a Bio-Plex Reader (Bio-Rad).

## Flow cytometry

Shaved back skin (~2 cm$^2$ containing both injection sites) was cut into small pieces and incubated for 1 h at 37 °C and constant shaking in 8 mL RPMI 1640 complete medium (R8758; Merck, Darmstadt, Germany) supplemented with 10% heat-inactivated FCS (Invitrogen, Waltham, Massachusetts, United States), 100 U/ml penicillin, 100 mg/ml streptomycin (both from Merck), 50 µM 2-mercaptoethanol (31350010; ThermoFisher), 0.25 mg/ml collagenase IV (C1889; Sigma-Aldrich), 0.166 mg/ml hyaluronidase (H3506; Sigma-Aldrich) and 0.1 mg/ml DNase I (11284932001; Merck). Skin-draining lymph nodes were incubated in for 20 min at 37 °C in 1 mL PBS supplemented with 2.4 mg/ml collagenase IV (C1889; Sigma-Aldrich) and 2.4 mg/ml DNase I (11284932001; Merck). After incubation, cells were filtered through a 100 µm cell strainer and incubated with surface markers for 20 min at 4 °C. Subsequently, cells were fixed according to the manufacturer's instructions (420801; Biolegend, San Diego, California, USA). Data was acquired using a CytoFLEX flow cytometer (Beckman-Coulter, Brea, California, USA) and analyzed with CytExpert (Beckman-Coulter). For the analysis of IFNγ production by skin cells, ear's skin was cut into small pieces and incubated for 45 min at 37 °C and constant shaking in 2 mL PBS supplemented with 2 mg/ml collagenase IV (C1889; Sigma-Aldrich), 0.5 mg/ml hyaluronidase (H3506; Sigma-Aldrich) and 0.1 mg/ml DNase I (11284932001; Merck). 2 × 10$^6$ skin cells were incubated with IL-2 (10 ng/mL), IL-12 (5 ng/µL, 210-12-A; PeproTech, London, UK) or 4 × 10$^6$ heat-killed *C. albicans* for 24 h at 37 °C. After incubation, cells were filtered through a 100 µm cell strainer. For the analysis of IFNγ production by splenocytes, the spleen was isolated, smashed through a 100 µm cell strainer, and red blood cell lysis was performed according to the manufacturer's instructions (R7757; Sigma-Aldrich). 1 × 10$^6$ splenocytes were incubated with IL-12 (5 ng/µL, 210-12-A; PeproTech, London, UK) or 2 × 10$^6$ heat-killed *C. albicans* for 24 h at 37 °C. After incubation, skin cells or splenocytes were stained and fixed as previously described. For intracellular staining, fixed cells were washed with Permeabilization Wash Buffer according to the manufacturer's instructions (421002; Biolegend) and incubated with intracellular markers for 30 min at RT. Data was acquired using a CytoFLEX flow cytometer (Beckman-Coulter, Brea, California, USA) and analyzed with CytExpert (Beckman-Coulter) or with a BD FACSCanto™ II flow Cytometer (BD, Franklin Lakes, New Jersey, USA) and analyzed with FlowJo (BD). The following antibodies, all from Thermofisher, were used: anti CD16/CD32 (14-0161-85), PE anti-F4/80 (12-4801-82), FITC anti F4/80 (11-4801-82), PE-610 anti-Ly-6G (61-9668-80), PerCp-Cy5.5 anti-Ly-6C (45-5932-80), PE-Cy7 anti-CD3e (25-0031-82), PB anti-CD3e (48-0033-82), APC-eF780 anti-CD3e (47-0031-82), SB600 anti-CD4 (63-0041-80), AF700 anti-CD8a (56-0081-82), PE anti-NK1.1 (12-5941-82), PB anti-NK1.1 (48-5941-82), PE-Cy7 anti-NK1.1 (25-5941-82), PE-Cy7 anti-CD19 (25-0193-82), AF700 anti-CD11c (56-0114-82), PB anti CD11b (48-0112-82), APC anti-CD11b (17-0112-82), AF488 anti-CD11b (53-0112-82), PE anti-CD90.2 (12-0902-81), SB780 anti-CD45 (78-0451-82), PerCp-Cy5.5 anti-IFNγ (45-7311-82), PE-Cy7 anti-IFNγ (25-7311-82). Additionally, AF700 anti-CD45 (103128; Biolegend), PE-Cy7 anti-TCRγδ (118124; Biolegend), APC anti-TCRβ (109212; Biolegend), PerCp-Vio700 anti-TCRβ (130-120-827; Miltenyi, Bergisch Gladbach, Germany) and two fixable viability dies (65-0865-18; 65-0863-14; Thermofisher) were used for the surface staining.

## Neutrophil depletion

On the day before the infection with *C. albicans*, mice were anesthetized as previously described for shaving the back skin. After shaving, mice were intraperitoneally injected with 0.5 µg/µL anti-Ly6G (Ab00295-2.0; Absolute antibody, Cleveland, United Kingdom) in 200 µL of PBS or mock-injected with PBS only. On the infection day, mice were anesthetized following the same protocol and intradermally infected as previously described. A drop of blood from the facial vein was collected before *C. albicans* infection to access efficient depletion. Two days after the infection, blood, and the back skin samples were collected and analyzed by flow cytometry to evaluate neutrophil depletion and the kidneys were harvested and processed for a CFU-assay as described previously.

## Cell sorting and RNA isolation for RNA-sequencing

For RNA-seq, the skin of infected mice was harvested and digested as described in the previous section. 30,000–70,000 skin-infiltrating myeloid cells (CD45$^+$CD3$^-$CD19$^-$NK1.1$^-$) were sorted using BD FACS Melody into 350 µL RLT buffer supplemented with β-mercaptoethanol and stored at −80 °C prior to RNA isolation. The following antibodies, all from Thermofisher, were used for FACS-sorting: anti CD16/CD32 (14-0161-85), PerCp-Cy5.5 anti-CD3e (45-0031-82), PerCp-Cy5.5 anti-NKp46 (46-3351-82), PerCp-Cy5.5 anti-CD19 (45-0193-82), and PE anti-CD45 (12-0451-83). Additionally, cells were stained with a fixable viability dye (65-0866-14; Thermosfisher). Total RNA was extracted using RNeasy Plus Micro Kit (74034; Qiagen) according manufacturer's instructions.

## NGS library preparation and raw data acquisition

For the preparation of NGS libraries from low-input samples we followed the Smart-seq2 protocol[68]. The subsequent library preparation from the amplified cDNA was performed using the Nextera XT DNA library prep kit (Illumina, San Diego, CA, USA). Library concentrations were quantified with the Qubit 2.0 Fluorometric Quantitation system (Life Technologies, Carlsbad, CA, USA) and the size distribution was assessed using the 2100 Bioanalyzer instrument (Agilent, Santa Clara, CA, USA). For sequencing, samples were diluted and pooled into NGS libraries in equimolar amounts.

Expression profiling libraries were sequenced on HiSeq 3000/4000 instruments (Illumina, San Diego, CA, USA) following a 50-base-pair, single-end recipe. Raw data acquisition (HiSeq Control Software, HCS, HD 3.4.0.38) and base calling (Real-Time Analysis Software, RTA, 2.7.7) was performed on-instrument, while the subsequent raw data processing of the instruments involved two custom programs based on Picard tools (2.19.2) (https://broadinstitute.github.io/picard/). In a

first step, base calls were converted into lane-specific, multiplexed, unaligned BAM files suitable for long-term archival (IlluminaBasecallsToMultiplexSam, 2.19.2-CeMM). In a second step, archive BAM files were demultiplexed into sample-specific, unaligned BAM files (IlluminaSamDemux, 2.19.2-CeMM).

## Differential gene expression and pathway analysis

Acquired NGS reads were aligned to the Genome Reference Consortium GRCm39 assembly using Spliced Transcripts Alignment to a Reference (STAR, v2.7.3a)[69], with the Ensembl transcript annotation from version e105 (December 2021) serving as reference transcriptome. Aligned reads were assigned to genomic features using featureCounts (v2.0.0)[70] and subsequently analyzed for differential expression via the RStudio (http://www.rstudio.com/) (v2023.03.1-446) Bioconductor (v3.15)[71] DESEq2 (v1.36.0)[72] package based on a negative binomial distributed model. Genes with low expression, as defined by a gene count logCPM below 1, were filtered out. An adaptive Log2-fold change shrinkage was applied using Empirical Bayes based on the Ashr (v2.2.63)[73] R package and significantly differentially expressed genes were extracted using the s-value threshold 0.05.

A heatmap displaying z-scores for differentially expressed genes between *Tyk2⁻/⁻* and *WT* or *Tyk2^K923E* and *WT* cells was generated using the Python (v3.9.13)[74] data visualization libraries matplotlib (v3.7.1)[75] and seaborn (v0.11.2)[76]. Respective z-scores were calculated using the Python sklearn (v1.2.1) library[77]. Pathway analysis of the differentially expressed genes was carried out using Reactome (v86)[78], an open-source, curated, and peer-reviewed pathway database. To maximize the chances of successful matches to the curated human pathways within Reactome, all non-human identifiers in the query were converted to their human equivalents.

## Statistical analysis

Statistical analysis was performed with GraphPad Prism 9. CFU, flow cytometry, and RT-qPCR data was log transformed for statistical analysis. One-way ANOVA followed by Tukey's multiple comparisons test was used when more than 2 groups were compared. Two-tailed Mann Whitney test was used for comparisons of 2 groups. CFU assay graphs show median values. All other graphs show the mean with standard error of the mean (SEM). For all graphs: $*p < 0.05$, $**p < 0.01$, $***p < 0.001$, $****p < 0.0001$.

## Reporting summary

Further information on research design is available in the Nature Portfolio Reporting Summary linked to this article.

# Data availability

All raw data in this study are available in the Source Data file, which has been uploaded with this manuscript. The RNA-seq data generated in this study have been deposited in the NCBI Gene Expression Omnibus (GEO) repository under accession code GSE278915. Source data are provided with this paper.

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

## Acknowledgements
We are grateful to Marion Bokor for histological sample preparation and immunohistochemical staining and Bettina Tutzer and Sara Lang for their help with in vivo experiments. This work was supported by the Austrian Science Fund (FWF) *doc.funds* doctoral program (*TissueHome*, DOC32-B28) to K.K. and B.S. and the SFB-F6101 to M.M. The work was furthermore supported in part by other FWF-funded projects to K.K. (ChromFunVir; P-32582-B08) and B.S. (P34286-B).

## Author contributions
S.M. and B.S. designed the study with input from K.K. S.M. designed and conducted the experiments, analyzed data, interpreted results, prepared the figures and wrote the manuscript together with B.S. and input from all co-authors; K.S. conducted experiments and analyzed data; V.B. conducted experiments; C.L. supervised and assisted in vivo experiments; S.T. was responsible for the pathological analysis of skin samples; H.K. and C.V. analyzed the RNA-seq data; C.V. helped with statistical analysis; I.T. provided technical input for the overall study; K.K., I.T., and M.M. provided critical input throughout the project. All authors read the manuscript and agreed on its publication.

## Competing interests
The authors declare no competing interests.
