## [Transparent Peer Review file · Nature Communications]

Lack of TYK2 Signaling Enhances Host Resistance to *Candida albicans* Skin Infection

Corresponding Author: Dr Birgit Strobl

Version 0:

Reviewer comments:

Reviewer #1

(Remarks to the Author)

Interesting study by Miranda et al to understand the role of TYK2 signaling in *Candida albicans* skin infection. The work is significant and novel as this is the first study to examine the role of this signaling pathway in fungal skin infection. However addressing the following concerns will improve the manuscript.

-Figures 1 and 2, utilizing flow cytometry (showing gating strategy and flow plots) in main figures would be much better to identify the accurate population and will increase the readability and quality

-Gating strategy to identify gamma delta T cells need to revised (PMID: 30870621). Using flow it is important to identify gamma delta T cell population in WT and TYK2^{-/-} mice infected with *C. albicans* will gain insights about local gamma delta T cells. It should be differentiated from epidermal dendritic T cells (PMID: 30870621)

- Although stimulation of spleen cells from WT and TYK2^{-/-} mice suggest IFN γ produced by gamma delta cells were increased. However direct evidence/experiments to support this conclusion is lacking. For example, antibody mediated T cell depletion could be considered to identify the T cell subsets that produces IFN γ in this model.

-To rule neutrophils play a role in *C. albicans* dissemination in WT and TYK2^{-/-} mice, neutrophil depletion experiment could be considered. In addition adding neutrophils, macrophages and monocytes data in Fig 4-6 would gain insights if phagocytes are altered in these knock out mice infected with *C. albicans*.

Reviewer #2

(Remarks to the Author)

This manuscript investigates the role of TYK2 in the immune response to *Candida albicans* using a cutaneous candidiasis model of dermal infection accompanied by systemic dissemination. The authors provide data indicating an unexpected detrimental role for TYK2 in cutaneous *C. albicans* infection. Using TYK2 KO mice models, the authors demonstrate that mice lacking TYK2 or its activity are more resistant to infection with *Candida albicans*, with reduced fungal burden in skin and more significantly in kidneys compared to WT controls. TYK2 deficiency leads to the formation of a necrotic neutrophil layer around the fungus, which they posit limits the spread of the fungus in the skin. TYK2 signaling in neutrophils is required for interferon-gamma-inducible gene expression, which they attribute to the antifungal activity of neutrophils. Inferential data from KO mice suggests that gamma-delta T cells are the principal source of IFN γ in this setting. Generally, the experiments are clear and mostly well controlled with exceptions noted below. There is novelty in that Tyk2's role in candidiasis is not well defined, and relevance as humans with TYK2 deficiencies have been described and a role for IFN-gamma in driving candidiasis in AIRE-deficiency (APECED) has been proposed but remains a topic of controversy. However, there major concerns that reduce enthusiasm, in particular regarding: (1) relevance of the mouse model utilized, since cutaneous candidiasis is not considered to be accompanied by fungal dissemination in humans, (2) weak and indirect demonstration of gamma-delta T cells as the source of IFN-gamma, (3) lack of true littermate controls, (4) a failure to interpret data in light of the epithelial-damaging activity of IFN γ in the setting of AIRE-deficiency (Lionakis' group, Break et al. Science 2021, PMID: 33446526).

Major Comments:

1. The model used here applies *C. albicans* at a high bolus (10^8 CFU) intradermally, which manifests as a nodule within the first 24 - 48 hours after the infection but within 2 days also exhibits dissemination to kidney, which appears to be sustained. This dissemination is absent in Tyk2-ko mice, which could have exciting implications for disease control. That said, the authors need to better justify to what extent this a relevant model for cutaneous *C. albicans* infections, as CMC patients rarely if ever exhibit systemic disease. Is this potentially an artefact of the high inoculum? The authors need to determine if the protection in Tyk2-ko and the Tyk2-kinase deficient mice is recapitulated in the more standard systemic candidiasis model induced via bloodstream infection with *Candida*.
2. Related to this concern with the model, authors see sustained weight loss of 5% in WT controls (not seen in the KO mice). Most models of cutaneous disease are not associated with weight loss, and it is notable here that the WT control mice are from a commercial vendor, and thus not exposed to the same microbiota as the experimental animals. It is essential that they repeat these studies with bona fide littermates (at minimum with the Tyk2 KO mice, though ideally also with the kinase-dead strain) to recapitulate the Tyk2 phenotype.
3. The link to gammadelta T cell functionality is unconvincing. The importance of gamma delta cells is largely inferred from the CD4-driven KO mice, but it is well known that there are compensatory increases in gamma-delta T cells when alphabeta T cells are missing and vice versa. Whether this compensatory effect is occurring needs to be shown. Moreover, the authors need to provide the primary flow cytometry data to show that IFN γ in each cell compartments is not dysregulated by depletion in other compartments. In Fig 7, the gamma-delta studies are conducted in vitro with splenocytes, not with skin-resident gamma-delta cells – this needs to be validated with dermal gamma-delta cells.
4. The authors largely overlook recent important findings showing that IFN γ drives oral mucosal *Candida* susceptibility (Break et al, Science, PMID: 33446526). The results presented here are supportive of that idea – namely, that IFN γ drives pathology on epithelial cells. This needs to be elaborated on in light of the present data. Related to this, STAT1-GOF mutations are one of the most common genetic associations with CMC in humans – if it is possible for the authors to test this model in a murine STAT1-GOF setting, that would strengthen the relevance, though if mice are not available this should at least be discussed.

Minor Comments:

1. Authors present a subset of the bioinformatics results “to maximize the chances of successful matches to the curated human pathways within Reactome, all non-human identifiers in the query were converted to their human equivalents.” (Fig 3). Given that this is acknowledged to be cherry-picked, the authors need to provide an additional set of heatmaps or volcano plots showing an unbiased list of the top 100 (or so) genes that are altered in this setting.
2. The authors overstate the roles of Tyk2 and Ifn γ in the skin manifestation of infection. While the difference in CFU count in the skin is reported to be statistically significant (or in some cases just a trend towards an increase), the difference overall in skin is modest or even negligible across knockout groups. Rather, data suggest that the more compelling roles of this pathway is to drive dissemination while only modestly increasing host resistance to intradermal infection. In this regard, it would be helpful to indicate the total CFU count (it can be hard to see on the log scale provided).
3. For all figures assessing lymphocytes and granulocyte population, the authors need to provide total numbers of each population in addition to percentage (always mandatory practice in flow cytometry), but especially important here as the knockout mice exhibit generally quite minor changes ($\pm 10\%$) in granulocyte and lymphocyte populations.
4. For all tissue imaging figures, staining should be quantified to support statements.
5. Figure 5B- “One representative out of 2 experiments is shown, n=1-2 (PBS) and n=5-7”, further repeats should be done to have a more robust dataset, or all experiments with total number of mice should be represented.
6. The authors do not provide a rationale for the use of heat killed *Candida* (Fig 7).

Reviewer #3

(Remarks to the Author)

This manuscript is well-written with an interesting finding for a role of TYK2 in control of *Candida albicans* at the skin site and for prevention of dissemination. The models and experiments appear appropriate for the scope of the study. I have the following comments:

Overall there was some concern that only 1 experiment is shown in the data. For microscopy, a representative image appears appropriate, but for other data, all should be included. There are many examples of this in the main figures and supplemental data.

Line 93: Provide reference for the lack of kinases activity for these mice.

Figure 1G: Can you show all the mice (both experiments) and statistics.

The supplementary data figure 3 for IFN should be in the main text. Maybe it could be swapped with the Figure 3E data?

Can the gamma delta cells be depleted in a manner to the alpha beta T cells?

Version 1:

Reviewer comments:

Reviewer #1

(Remarks to the Author)

Findings are interesting and will be significant addition to the field. Results support the conclusion.

Reviewer #2

(Remarks to the Author)

The authors have addressed concerns in a satisfactory manner.

Reviewer #3

(Remarks to the Author)

The authors have addressed my comments

October 7, 2024 - Point-by-point reply

RE: Nature Communications manuscript NCOMMS-24-06128

“Lack of TYK2 Signaling Enhances Host Resistance to *Candida albicans* Skin Infection”

REVIEWER COMMENTS

Reviewer #1 (Remarks to the Author):

Interesting study by Miranda et al to understand the role of TYK2 signaling in *Candida albicans* skin infection. The work is significant and novel as this is the first study to examine the role of this signaling pathway in fungal skin infection. However addressing the following concerns will improve the manuscript.

We thank the reviewer for the thorough and positive evaluation of our manuscript and for the helpful and constructive comments. Responses to all comments are indicated in blue, changes to the manuscript are highlighted in yellow.

-Figures 1 and 2, utilizing flow cytometry (showing gating strategy and flow plots) in main figures would be much better to identify the accurate population and will increase the readability and quality

We agree that showing gating strategies and flow plots for all flow cytometry data will improve the overall clarity.

Figure 1D (blood immune cells): data are from a blood analyzer (Vet ABC) and show the % of granulocytes and lymphocytes out of white blood cells (WBC). We included the total number of cells in the **Supplementary Fig. S1** of the revised manuscript.

Flow cytometry data and plots for blood neutrophils of *C. albicans* infected *WT* and *Tyk2^{-/-}* mice are shown in **Supplementary Fig. S3** in the revised manuscript.

Figure 2 (skin-infiltrating cells): as suggested by the reviewer, gating strategies and flow plots are included in the revised manuscript.

In addition, gating strategies and representative flow plots were added for all flow cytometry experiments in the revised manuscript (**Figs. 3b, 3c, S3a, S3b, 5g, 6f, 7j, S7d, S7f, 8b, S8a, S8b, S8d**).

-Gating strategy to identify gamma delta T cells need to revised (PMID: 30870621). Using flow it is important to identify gamma delta T cell population in *WT* and *TYK2^{-/-}* mice infected with *C. albicans* will gain insights about local gamma delta T cells. It should be differentiated from epidermal dendritic T cells (PMID: 30870621).

We thank the reviewer for the insightful comment. Our data were generated using $\gamma\delta$ T cells from spleens lacking epidermal dendritic T cells. However, we agree that it is important to analyze $\gamma\delta$ T cells from the skin. As suggested by the reviewer, we repeated the experiments following the protocols used in PMID: 30870621 to discriminate local $\gamma\delta$ T cells from epidermal dendritic T cells. Similar to our finding with splenic $\gamma\delta$ T cells, we show that skin $\gamma\delta$ T cells produce $IFN\gamma$ when challenged with heat-killed *C. albicans* in a TYK2-dependent manner. Data are shown in **Fig. 8b-d** and gating strategies are shown in **Supplementary Fig. S8a** of the revised manuscript.

- Although stimulation of spleen cells from WT and TYK2^{-/-} mice suggest IFN γ produced by gamma delta cells were increased. However direct evidence/experiments to support this conclusion is lacking. For example, antibody mediated T cell depletion could be considered to identify the T cell subsets that produces IFN γ in this model.

We appreciate this point and we agree that depletion of $\gamma\delta$ T cells would unequivocally show that this T cell subset produces IFN γ in our model. However, unfortunately, the tools are missing. Treatment with an anti- $\gamma\delta$ TCR antibody *in vivo* results in downregulation of the $\gamma\delta$ TCR but not depletion of $\gamma\delta$ T cells (PMID: 19130484; PMID: 30463878). Given the rather low abundance of $\gamma\delta$ T cells even in infected skin tissues, the experiment would be technically very challenging. Of note, to address the reviewer's comment, we included this limitation of our study in the discussion of the revised manuscript (**page 13, line 378**).

-To rule neutrophils play a role in *C. albicans* dissemination in WT and TYK2^{-/-} mice, neutrophil depletion experiment could be considered. In addition adding neutrophils, macrophages and monocytes data in Fig 4-6 would gain insights if phagocytes are altered in these knock out mice infected with *C. albicans*.

We thank the reviewer for the constructive and valid point. As suggested, we performed neutrophil-depletion experiments in WT and *Tyk2*^{-/-} mice. Indeed, neutrophil depletion increased *C. albicans* invasion and dissemination into kidneys in *Tyk2*^{-/-} mice to levels observed in WT mice, strongly supporting the notion that neutrophils prevent fungal dissemination in *Tyk2*^{-/-} mice. The new data are shown in the new **Fig. 3** and Supplementary **Fig. S3** in the revised manuscript.

Further, we also analyzed skin-infiltrating macrophages, monocytes and neutrophils by flow cytometry in *C. albicans*-infected (i) WT and *Ifngr1*^{-/-} mice (Figure 4), (ii) WT and *Rag2*^{-/-} mice (Figure 5) and (iii) *Tyk2*^{fl/fl} and *Tyk2* ^{Δ T} mice (Figure 6). We did not observe differences between any of these knock-out and control mice, indicating that none of these phagocyte populations are altered in gene-targeted mice. Data are shown in **Figs. 5-7** and **Supplementary Figs. S5-S7** of the revised manuscript.

Reviewer #2 (Remarks to the Author):

This manuscript investigates the role of TYK2 in the immune response to *Candida albicans* using a cutaneous candidiasis model of dermal infection accompanied by systemic dissemination. The authors provide data indicating an unexpected detrimental role for TYK2 in cutaneous *C. albicans* infection. Using TYK2 KO mice models, the authors demonstrate that mice lacking TYK2 or its activity are more resistant to infection with *Candida albicans*, with reduced fungal burden in skin and more significantly in kidneys compared to WT controls. TYK2 deficiency leads to the formation of a necrotic neutrophil layer around the fungus, which they posit limits the spread of the fungus in the skin. TYK2 signaling in neutrophils is required for interferon-gamma-inducible gene expression, which they attribute to the antifungal activity of neutrophils. Inferential data from KO mice suggests that gamma-delta T cells are the principal source of IFN γ in this setting. Generally, the experiments are clear and mostly well controlled with exceptions noted below. There is novelty in that *Tyk2*'s role in candidiasis is not well defined, and relevance as humans with TYK2

deficiencies have been described and a role for IFN-gamma in driving candidiasis in AIRE-deficiency (APECED) has been proposed but remains a topic of controversy. However, there major concerns that reduce enthusiasm, in particular regarding: (1) relevance of the mouse model utilized, since cutaneous candidiasis is not considered to be accompanied by fungal dissemination in humans, (2) weak and indirect demonstration of gamma-delta T cells as the source of IFN-gamma, (3) lack of true littermate controls, (4) a failure to interpret data in light of the epithelial-damaging activity of IFNgamma in the setting of AIRE-deficiency (Lionakis' group, Break et al. Science 2021, PMID: 3 3446526).

We thank the reviewer for the thorough evaluation of our manuscript, and for the helpful and constructive comments. Responses to all comments are indicated in blue, changes to the manuscript are highlighted in yellow.

Major Comments:

1. The model used here applies *C. albicans* at a high bolus (10^8 CFU) intradermally, which manifests as a nodule within the first 24 - 48 hours after the infection but within 2 days also exhibits dissemination to kidney, which appears to be sustained. This dissemination is absent in Tyk2-ko mice, which could have exciting implications for disease control. That said, the authors need to better justify to what extent this a relevant model for cutaneous *C. albicans* infections, as CMC patients rarely if ever exhibit systemic disease. Is this potentially an artefact of the high inoculum? The authors need to determine if the protection in Tyk2-ko and the Tyk2-kinase deficient mice is recapitulated in the more standard systemic candidiasis model induced via bloodstream infection with *Candida*.

This insightful point is well taken. We discuss the experimental model used and, based on a comment by reviewer #2, applicable limitations concerning translation to human patients in the revised manuscript (**page 13, line 364ff**). In addition, we performed a pilot experiment (n=4-6 per genotype) with standard systemic infections by *C. albicans*. We observed organ-specific effects of TYK2 (**see attached Figure for inspection by reviewers**), such as decreased CFUs in the liver at day 3 p.i. This suggests that a protective role of TYK2 is not solely confined to the skin infection model. Surprisingly, we also found no differences in CFUs in the kidneys but transiently increased CFUs in the lung. Although these data need confirmation, they suggest that the high-dose intradermal infection model does not simply mimic standard systemic infections. Moreover, our preliminary data indicate organ-specific roles of TYK2 and kinase-inactive TYK2 in the immune defense against blood-borne *C. albicans* infections. These new data are remarkable and warrant further exploration, but we feel that characterizing the roles of TYK2 in tissue-specific setting would go beyond the scope of this work.

2. Related to this concern with the model, authors see sustained weight loss of 5% in WT controls (not seen in the KO mice). Most models of cutaneous disease are not associated with weight loss, and it is notable here that the WT control mice are from a commercial vendor, and thus not exposed to the same microbiota as the experimental animals. It is essential that they repeat these studies with bona fide littermates (at minimum with the Tyk2 KO mice, though ideally also with the kinase-dead strain) to recapitulate the Tyk2 phenotype.

We apologize for the confusion and lack of clarity. *WT* mice were bred at the University of Veterinary Medicine in the same facility as all gene-targeted mice (**page 14, line 388**). We did not use mice from a commercial vendor, but we mentioned the original source (Janvier Labs) in the original manuscript. To avoid confusion, we provide the background of the mice (C57BL/6N) and relevant references, while we removed the original source from the method section of the revised manuscript.

3. The link to gammadelta T cell functionality is unconvincing. The importance of gamma delta cells is largely inferred from the CD4-driven KO mice, but it is well known that there are compensatory increases in gamma-delta T cells when alphabeta T cells are missing and vice versa. Whether this compensatory effect is occurring needs to be shown. Moreover, the authors need to provide the primary flow cytometry data to show that IFN γ in each cell compartments is not dysregulated by depletion in other compartments. In Fig 7, the gamma-delta studies are conducted in vitro with splenocytes, not with skin-resident gamma-delta cells – this needs to be validated with dermal gamma-delta cells.

The point is indeed well-taken. Hence, we performed flow cytometry experiments and found that the absence of TYK2 in $\alpha\beta$ T cells does not increase the number of $\gamma\delta$ T cells in the skin or in skin-draining lymph nodes. The new data are shown in **Supplementary Fig. S7d-e** of the revised manuscript. Of note, we were unable to detect IFN γ by flow cytometry in any of the cell types. Thus, direct evidence for $\gamma\delta$ T cells being the major producers of IFN γ is still missing. We included this limitation of our study in the discussion of the revised manuscript (**page 13, line 378**).

For data in Figure 7, we followed the reviewer's suggestion and repeated the experiments with skin-resident $\gamma\delta$ T cells. We used the gating strategy suggested by reviewer #1 to exclude dendritic epidermal T cells (DETC) from our analysis. Similar to our findings with splenic $\gamma\delta$ T cells, we found a TYK2-dependent IFN γ production by skin-derived $\gamma\delta$ T cells in response to IL-12 and heat-killed *C. albicans*. The new data and representative flow plots are shown in **Fig. 8** and the gating strategy in **Supplementary Fig. S8a** of the revised manuscript.

4. The authors largely overlook recent important findings showing that IFN γ drives oral mucosal *Candidia* susceptibility (Break et al, Science, PMID: 33446526). The results presented here are supportive of that idea – namely, that IFN γ drives pathology on epithelial cells. This needs to be elaborated on in light of the present data. Related to this, STAT1-GOF mutations are one of the most common genetic associations with CMC in humans – if it is possible for the authors to test this model in a murine STAT1-GOF setting, that would strengthen the relevance, though if mice are not available this should at least be discussed.

We apologize for the omission and we thank reviewer #2 for noting it. We discuss our findings in the light of this report (by Break et al.) in our revised manuscript (**page 12, line 330**). As STAT1-GOF mice are not available to us, we followed the suggestion and discuss STAT1-GOF and CMC in the revised manuscript (**page 12, line 338**).

Minor Comments:

1. Authors present a subset of the bioinformatics results “to maximize the chances of successful matches to the curated human pathways within Reactome, all non-human identifiers in the query were converted to their human equivalents.” (Fig 3). Given that this is acknowledged to be cherry-picked, the authors need to provide an

additional set of heatmaps or volcano plots showing an unbiased list of the top 100 (or so) genes that are altered in this setting.

We apologize for being a bit quick on this matter. Hence, we provide (i) volcano plots and (ii) an unbiased list of all differentially expressed genes ($p < 0.05$) in an amended **Fig. 4b**, **Supplementary Fig. S4a** and **Supplementary Data** of the revised manuscript. The heatmap in **Fig. 4c** of the revised manuscript also provides an unbiased list of all genes changing expression levels by more than 2-fold between *WT* and *Tyk2*^{-/-} and/or *WT* and *Tyk2*^{K923E} cells ($p < 0.05$, absolute $\log_{2}FC > 1$).

2. The authors overstate the roles of *Tyk2* and *Ifng* in the skin manifestation of infection. While the difference in CFU count in the skin is reported to be statistically significant (or in some cases just a trend towards an increase), the difference overall in skin is modest or even negligible across knockout groups. Rather, data suggest that the more compelling role of this pathway is to drive dissemination while only modestly increasing host resistance to intradermal infection. In this regard, it would be helpful to indicate the total CFU count (it can be hard to see on the log scale provided).

We agree that the differences in CFU counts in the skin between *WT* and *Tyk2*^{-/-} mice are modest and that the most compelling role of *TYK2* is to allow for systemic dissemination to kidneys. To clarify, we have revised the respective sentences in the abstract, results (**page 5, line 107**) and discussion (**page 13, line 382**) of the revised manuscript. In addition, we show the total CFU counts (per skin biopsies and kidneys) across all genotypes in the **Supplementary Fig. S1a, S1b, S5a, S5b, S6b, S6g, S7a and S7b** of the revised manuscript.

3. For all figures assessing lymphocytes and granulocyte population, the authors need to provide total numbers of each population in addition to percentage (always mandatory practice in flow cytometry), but especially important here as the knockout mice exhibit generally quite minor changes ($\pm 10\%$) in granulocyte and lymphocyte populations.

We included the total cell numbers of all cell populations identified by flow cytometry in the revised manuscript (**Supplementary Figs. S2, S3, S5, S6 and S7**).

4. For all tissue imaging figures, staining should be quantified to support statements. To improve clarity, and based on comments by reviewer #1, we performed flow cytometry experiments to further support the histology/microscopy data generated by tissue imaging. The new data are presented in **Fig. 2c, 5g and 7j and related Supplementary Figs.** in the revised manuscript.

5. Figure 5B- “One representative out of 2 experiments is shown, $n=1-2$ (PBS) and $n=5-7$ ”, further repeats should be done to have a more robust dataset, or all experiments with total number of mice should be represented.

We agree with the reviewer and we now show pooled data rather than one representative experiment in the revised manuscript. For the data sets showing high variation between experiments, we show data from independent experiments separately in the revised manuscript (**Supplementary Fig. S1a and S8e**). Where needed or appropriate, we performed repeats to obtain robust data sets (**Fig. 1g**).

6. The authors do not provide a rationale for the use of heat killed Candida (Fig 7). Heat-killed Candida was used because live *C. albicans* would quickly overgrow and interfere with T cell functionality in *in vitro* cultures. Heat-killed Candida do not release extracellular proteases yet are strong stimulators of immune responses owing to the increased exposure of cell wall glucan molecules.

Reviewer #3 (Remarks to the Author):

This manuscript is well-written with an interesting finding for a role of TYK2 in control of *Candida albicans* at the skin site and for prevention of dissemination. The models and experiments appear appropriate for the scope of the study. I have the following comments:

We thank the reviewer for the thorough and positive evaluation of our manuscript and for the helpful and constructive feedback. Responses to all comments are indicated in blue, changes to the manuscript are highlighted in yellow.

Overall there was some concern that only 1 experiment is shown in the data. For microscopy, a representative image appears appropriate, but for other data, all should be included. There are many examples of this in the main figures and supplemental data.

We agree with the reviewer and we apologize. We show the pooled data rather than one representative experiments in the revised manuscript. For the data sets showing high variation between experiments, we show data from independent experiments separately in the revised manuscript (**Supplementary Fig. S1a and S8e**).

Line 93: Provide reference for the lack of kinases activity for these mice.

We added the reference (Prchal-Murphy et al., 2012; PMID: 22723949) **in line 93** of the revised manuscript.

Figure 1G: Can you show all the mice (both experiments) and statistics.

We show all the mice and the relevant statistics in **Fig. 1g** of the revised manuscript.

The supplementary data figure 3 for IFN should be in the main text. Maybe it could be swapped with the Figure 3E data?

We agree and appreciated this constructive suggestion, Thus, we swapped the Supplementary Fig. 3 with Figure 3E (**Supplementary Fig. 4b and Fig. 4e, f** in the revised manuscript).

Can the gamma delta cells be depleted in a manner to the alpha beta T cells?

Unfortunately, it is challenging to impossible to deplete $\gamma\delta$ T in a similar way as $\alpha\beta$ T cells (see also our response to reviewer #1). Although anti- $\gamma\delta$ TCR antibody treatment *in vivo* results in downregulation of $\gamma\delta$ TCR, a depletion of $\gamma\delta$ T cells cannot be achieved (PMID: 19130484; PMID: 30463878). Therefore, we state in the revised manuscript that a direct evidence for $\gamma\delta$ T cells being a major source of IFN γ is missing in the paragraph addressing the limitations of our study (**page 13, line 378**).

Figure for inspection by reviewers

Legend. **a** WT, *Tyk2*^{-/-} and *Tyk2*^{K923E} mice were intravenously (i.v) injected with 6x10⁴ CFU of *C. albicans* per 21.5g of body weight. **b** Body weight was measured overtime and is shown as percentage relative to the weight on the infection day (set as 100%). **c** Fungal load in the spleen, kidneys, liver, brain and lungs was measured on days 1, 3 and day 7 p.i. One experiment is shown shown. Mean values \pm SEM are given; n=4-6/genotype/time-point; **p < 0.01; n: biological replicates; Statistical significance is only given for the comparison between the genotypes). Dunnett's multiple comparisons test was used.